# Robust Model Selection of Gaussian Graphical Models

**Abrar Zahin**                                                                                      *azahin@asu.edu*
*School of Electrical, Computer & Energy Engineering*
*Arizona State University*

**Rajasekhar Anguluri**\*                                                                      *rajangul@umbc.edu*
*Department of Computer Science & Electrical Engineering*
*University of Maryland, Baltimore County*

**Lalitha Sankar**                                                                                  *lsankar@asu.edu*
*School of Electrical, Computer & Energy Engineering*
*Arizona State University*

**Oliver Kosut**                                                                                      *okosut@asu.edu*
*School of Electrical, Computer & Energy Engineering*
*Arizona State University*

**Gautam Dasarathy**                                                                          *gautamd@asu.edu*
*School of Electrical, Computer & Energy Engineering*
*Arizona State University*

**Reviewed on OpenReview:** *https://openreview.net/forum?id=AIby9MQXbu*

## Abstract

In Gaussian graphical model selection, noise-corrupted samples present significant challenges. It is known that even minimal amounts of noise can obscure the underlying structure, leading to fundamental identifiability issues. A recent line of work addressing this "robust model selection" problem narrows its focus to tree-structured graphical models. Even within this specific class of models, exact structure recovery is shown to be impossible. However, several algorithms have been developed that are known to provably recover the underlying tree-structure up to an (unavoidable) equivalence class.

In this paper, we extend these results beyond tree-structured graphs. We first characterize the equivalence class up to which general graphs can be recovered in the presence of noise. Despite the inherent ambiguity (which we prove is unavoidable), the structure that can be recovered reveals local clustering information and global connectivity patterns in the underlying model. Such information is useful in a range of real-world problems, including power grids, social networks, protein-protein interactions, and neural structures. We then propose an algorithm which provably recovers the underlying graph up to the identified ambiguity. We further provide finite sample guarantees in the high-dimensional regime for our algorithm and validate our results through numerical simulations.

## 1 Introduction

Probabilistic graphical models have emerged as a powerful and flexible formalism for expressing and leveraging relationships among entities in large interacting systems (Lauritzen, 1996). They have found application in

---

\*The work was primarily done while R.A. was at Arizona State University.

a range of areas including signal processing (Kim & Smaragdis, 2013; Ott & Stoop, 2006; Murphy et al., 2013), power systems (Anguluri et al., 2022; Deka et al., 2020; 2015), (phylo)genomics (Zuo et al., 2017; Dasarathy et al., 2014; 2022), and neuroscience (Bullmore & Bassett, 2011; Vinci et al., 2019). Gaussian graphical models are an important subclass of graphical models and are the main focus of this paper; our techniques do apply more broadly, as discussed in Section 7.

c In several applications, we do not know the underlying graph structure, and the goal is to learn this from data — a problem dubbed graphical model selection. This is important because the graph structure provides a succinct representation of the complex multivariate distribution and can reveal important relationships among the underlying variables. See, e.g., Drton & Maathuis (2017); Maathuis et al. (2018) and references therein for more on this problem. Here, we focus on a relatively new but important task where samples from the underlying distribution are corrupted by independent noise with unknown variances. This occurs in a wide variety of applications where sensor data or experimental measurements suffer from statistical uncertainty or measurement noise. In these situations, we refer to the task of graph structure learning as *robust model selection*.

This problem was recently considered by Katiyar et al. (2019) and a line of follow-up work (Katiyar et al., 2020; Casanellas et al., 2024; Tandon et al., 2021) who show that unfortunately the conditional independence structure of the underlying distribution can be completely lost in general under such corruption; see Section 3.1 for more on this. The authors show that even in the often tractable case of tree-structured graphical models, one can only identify the structure up to an equivalent class. In fact, the assumption that the underlying uncorrupted graphical model has a tree structure is critical to the techniques of this line of work. As Casanellas et al. (2024) astutely observe, when the random vector associated with the true underlying graph is corrupted with independent but non-identical additive noise, the robust estimation problem reduces to a latent tree structure learning problem. We improve on the algorithmic and theoretical results of this line of work by considering the robust model selection problem for general graphs. Our main contributions are summarized below.

- We establish a fundamental identifiability result (c.f. Theorem 3.1) for general graphs in the robust Gaussian graphical model selection problem. This confirms that the identifiability problem is exacerbated if one considers more general graphs. More importantly, this also generalizes the identifiability results from earlier lines of work and identifies an equivalence class up to which one may hope to recover the underlying structure.

- We devise a novel algorithm, called NoMAD (for Noisy Model selection based on Ancestor Discovery), that tackles the robust model selection problem for *general* graphs extending the results of (Katiyar et al., 2019; 2020; Casanellas et al., 2024; Tandon et al., 2021). Our algorithm is based on a novel "ancestor discovery" procedure (see Section 3.4) that we expect to be of independent interest. It is worth observing that the tree-based algorithms previously proposed fail, often catastrophically, when there are loops in the underlying graph.

- We show that NoMAD provably recovers the underlying graph up to a small equivalence class (c.f. Theorem 4.2) and establish sample complexity results (c.f. Theorem 5.3) for partial structure recovery in the high-dimensional regime.

- We also show the efficacy of our algorithm through experiments on synthetic and realistic network structures.

## 2 Related Work

Several lines of research have tackled the problem of robust estimation of high-dimensional graphical models under corruption. This includes graphical modeling with missing data, outliers, or bounded noise, see, for instance, Loh & Wainwright (2011); Chen et al. (2013); Wang & Lin (2014); Nguyen et al. (2022) and references therein. For the missing data problem, several other algorithms have been proposed for estimating mean values and covariance matrices from the incomplete dataset available to the learner Little & Rubin (2019); Schneider (2001); Lounici (2014). Zheng & Allen (2024) considered a variant of the missing value problem where instead of missing values, the measurements are irregular; that is, different vertex pairs have

vastly different sample sizes. Vinci et al. (2019); Chang et al. (2023); Dasarathy (2019) explored the situations where one is only able to obtain samples from subsets of variables, possibly missing joint observations from several pairs. Sun & Li (2012) and Yang & Lozano (2015) proposed algorithms for handling the outliers. There is another line of work that treats this problem using the error-in-variables lens (see the books and papers Hwang (1986); Carroll et al. (1995); Iturria et al. (1999); Xu & You (2007) and references therein). For the problem of model selection from bounded noisy measurements, see Wang & Lin (2014); Öllerer & Croux (2015); Loh & Tan (2018); Chen et al. (2015).

However, these papers do not consider the setting of unknown additive noise and the corresponding implications on the conditional independence structure. Recently, Nikolakakis et al. (2019) considered recovering forest-structured graphical models assuming that noise distribution across all vertices is identical. In contrast, our setting allows for unknown and non-identical noise. The robust model selection problem, as considered here, had not been adequately addressed even for the tree-structured graphical models until the recent work by Katiyar et al. (2019; 2020) who showed that the structure recovery in the presence of unknown noise is possible only up to an equivalence class. These studies also proposed algorithms to recover the correct equivalence class from noisy samples. Using information-theoretic methods, Tandon et al. (2021) improved the sample complexity result of Katiyar et al. (2020); Nikolakakis et al. (2019) and provided a more statistically robust algorithm for partial tree recovery. Finally, Zhang & Tan (2021) studied the structure recovery problem under noise when the nodes of the GGM are vector-valued random variables. However, these results are limited to tree-structured graphical models, as they inherently use the additive distance metric property to learn the full (or partial) structure. However, the additive distance metric does not hold for general structures (see more on Section 3.3). The results of this paper significantly extend this line of work, and are applicable to general graphs.

## 3 Preliminaries and Problem Statement

**Graph theory.** Let $G = (V, E)$ be an undirected graph on vertex set $V$ (with cardinality $p$) and edge set $E \subset \binom{V}{2}$. For a vertex $v \in V$, let $N_v \triangleq \{u \in V : \{u, v\} \in E\}$ be the neighborhood of the vertex $v \in V$ and *degree* $\deg(v)$ be the size of $N_v$. A vertex $v$ is said to be *leaf* if $\deg(v) = 1$. A *subgraph* of $G$ is any graph whose vertices and edges are subsets of those of $G$. For $V' \subseteq V$ the *induced subgraph* $G(V')$ has the vertex set $V'$ and the edge set $E' = \{\{u, v\} \in E : u, v \in V'\}$. A *path* between the vertices $u, v$ is a sequence of distinct vertices $v_1 = u, v_2, \ldots, v_k = v$ such that $\{v_i, v_{i+1}\} \in E$, for $1 \leq i < k$. We let $\mathcal{P}_{uv}$ denote the set of all paths between $u$ and $v$. If $\mathcal{P}_{uv}$ is not empty, we say $u$ and $v$ are connected. The graph $G$ is connected if every pair of vertices in $G$ is connected. A set $S \subseteq V$ separates two disjoint subsets $A, B \subseteq V$ if any path from $A$ to $B$ contains a vertex in $S$. We denote this separation as $A \perp\!\!\!\perp B \mid S$. The resemblance of this notation to that of the conditional independence of random variables in the graphical model will be made clear later.

**Gaussian graphical models.** Let $\mathbf{X} = (X_1, X_2, \ldots, X_p) \in \mathbb{R}^p$ be a zero-mean Gaussian random vector with a covariance matrix $\Sigma \in \mathbb{R}^{p \times p}$. Compactly, $\mathbf{X} \sim \mathcal{N}(\mathbf{0}, \Sigma)$, where $\mathbf{0}$ is the $p$-dimensional vector of all zeros. Let $G = ([p], E)$ be a graph on the vertex set $[p] \triangleq \{1, 2, \ldots, p\}$ representing the coordinates of $\mathbf{X}$. Let $K \triangleq \Sigma^{-1}$ is called the *precision matrix* of $\mathbf{X}$. The distribution of $\mathbf{X}$ is said to be a *Gaussian graphical model* (or equivalently, Markov) with respect to $G$ if $K_{ij} = 0$ for all $\{i, j\} \notin E$ [1]. In other words, for any $\{i, j\} \notin E$, $X_i$ and $X_j$ are conditionally independent given all the other coordinates of $\mathbf{X}$ (see Lauritzen (1996) for more details). In the sequel, we will use a generic set $V$ to denote the vertex set of our graph with the understanding that every element in $V$ is uniquely mapped to a coordinate of the corresponding random vector $\mathbf{X}$. For a vertex $v \in V$, with a slight abuse of notation, we write $X_v$ to denote the corresponding coordinate of $\mathbf{X}$. Similarly, we write $\Sigma_{uv}$ to mean the covariance between $X_u$ and $X_v$.

### 3.1 The Robust Model Selection Problem

In this paper, we consider a variant of the model selection problem, which we refer to as *robust model selection*. Formally, let $\mathbf{X} \sim \mathcal{N}(\mathbf{0}, \Sigma)$ be a Gaussian graphical model with respect to an unknown graph $G = (V, E)$. In the robust model selection problem, the goal is to estimate the edge set $E$ (or equivalently the sparsity

---

[1]Hence, the sparsity pattern of $K$ is represented by the edge set of $G$

pattern of $K = \Sigma^{-1}$) when one only has access to noisy samples of $\mathbf{X}$. That is, we suppose we have access to the corrupted version $Y$ of the underlying random vector $X$ such that $\mathbf{Y} = \mathbf{X} + \mathbf{Z}$, where the noise $\mathbf{Z} \sim \mathcal{N}(\mathbf{0}, D)$ is independent of $\mathbf{X}$ and $D$ is assumed to be diagonal with possibly distinct and even zero entries. In other words, the noise is assumed to be independent and heteroscedastic while potentially allowing for some coordinates of $\mathbf{X}$ to be observed uncorrupted. Observe that $\mathbf{Y} \sim \mathcal{N}(\mathbf{0}, \Sigma^o)$, where $\Sigma^o \triangleq \Sigma + D$. Indeed, $D$ is assumed to be unknown.

Unfortunately, such corruption can completely obliterate the conditional independence structure of $\mathbf{X}$. For instance, suppose that $D = e_j e_j^T$, where $e_j$ is a vector of zeros except in the $j^{th}$ entry where it is one. By the Sherman-Morrison identity (see e.g., Horn & Johnson, 2012), we have $(\Sigma^o)^{-1} = K - cK e_j e_j^\top K$ for some $c \geq 0$. The term $K e_j e_j^\top K$ can be dense in general and, hence, can fully distort the sparsity of $K$ (the conditional independence structure of $\mathbf{X}$).

In view of the above example, the robust model selection problem appears intractable. As outlined in Section 1, recent studies show that this problem is partially tractable for tree-structured graphs. Notably, as Casanellas et al. (2024) astutely observes, one can reduce the problem of robust structure estimation of trees to the problem of learning the structure of latent tree graphical models, which enjoy several efficient algorithms and rich theoretical results (see, e.g., Choi et al. (2011); Erdös et al. (1999); Semple & Steel (2003); Dasarathy et al. (2014)). To see this, suppose that $\mathbf{X}$ is Markov according to a graph $G = (V = [p], E)$ that is tree-structured. Let the *joint graph* $G^{\text{J}}$ denote the graph obtained by creating a copy of each node in $G$ and linking the copies to their counterparts in $G$. Formally, we define $G^{\text{J}} = (V^{\text{J}}, E^{\text{J}})$, where $V^{\text{J}} \triangleq V \cup \{1^e, 2^e, \ldots, p^e\}$ and $E^{\text{J}} \triangleq E \cup \{\{i, i^e\} : i \in [p]\}$. In subsequent sections, for any vertex subset $B \subseteq 2^V$, we let $B^e$ denote the vertices associated with the noisy observations from $B$ and call it the *noisy counterpart* of $B$.

Notice that with this definition, if we associate the coordinates of $\mathbf{Y}$ to the newly added leaf vertices, the concatenated random vector $[\mathbf{X}; \mathbf{Y}]$, obtained by stacking $\mathbf{X}$ on top of $\mathbf{Y}$, is Markov according to $G^{\text{J}}$. Casanellas et al. (2024) then uses the fact that when given samples of $\mathbf{Y}$, one can reconstruct a reduced latent tree representation of $G^{\text{J}}$, which in turn can be used to infer an equivalence class of trees that contains the true tree $G$. Indeed, the equivalence class thus obtained is the same one identified by Katiyar et al. (2019; 2020). We next state our general identifiability result after introducing some more graph-theoretic concepts.

## 3.2 An Identifiability Result

A connected graph $G$ is said to be *biconnected* if at least 2 vertices need to be removed to disconnect the graph. A subgraph $H$ of $G$ is said to be a *biconnected component* if it is a maximal biconnected subgraph of $G$. That is, $H$ is not a strict subgraph of any other biconnected subgraph of $G$. The vertex set of such a biconnected component will be referred to as a *block*. A block is *non-trivial* if it has more than two vertices. For example, in Fig. 1a, the vertex set $\{1, 2, 3, 4\}$ and $B_1 \cup \{6, 8\}$ are non-trivial blocks where $B_1$ is an arbitrary set of vertices such that the subgraph on $B_1 \cup \{6, 8\}$ is a biconnected component, whereas, the set $\{10, 8\}$ is a trivial block. In what follows, we will often be interested in the vertices of such non-trivial blocks and toward this we write $\mathcal{B}^{\text{NT}}$ to denote the set of all vertex sets of non-trivial blocks in $G$. From these definitions, it follows that trees (which are cycle free) do not have any non-trivial blocks. It also follows that two blocks can share at most one vertex; we refer to such shared vertices as *cut* vertices. In Fig. 1a, the vertices 4 and 10 are cut vertices. The vertices in $\mathcal{B}^{\text{NT}}$ which are not cut vertices are referred to as *non-cut* vertices. In Fig. 1a, the vertex 1 is a non-cut vertex.

With these definitions, we now introduce a novel representation for a graph $G$ that will be crucial to stating our results. This representation is a tree-structured graph $\mathcal{T}_{\text{AST}}(G)$, which we call the *articulated set tree* of $G$, whose vertices correspond to (a) non-trivial blocks in $G$, and (b) vertices in $G$ that are not a member of any non-trivial blocks. Vertices in this tree-structured representation are connected by edges if the corresponding sets in $G$ either share a vertex or are connected by a single edge. The vertices in the original graph that are responsible for the edges in the representation are called articulation points[2]. We formally define this below. For example, for $G$ in Fig. 1a, the articulated set tree is illustrated in Fig. 1c, where the sets $\{1, 2, 3, 4\}$ and

---

[2]Articulation (points) vertices are cut vertices that separate non-trivial blocks from the rest of the graph.

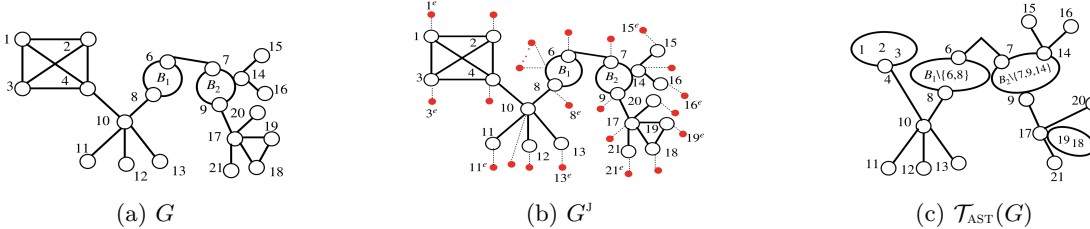

(a) $G$       (b) $G^{\text{J}}$       (c) $\mathcal{T}_{\text{AST}}(G)$

Figure 1: (a) a true underlying graph where both $B_1 \cup \{6,8\}$ ($B_2 \cup \{7,9\}$) are non-trivial blocks where $B_1$ ($B_2$) is an arbitrary set of vertices such that the subgraph on $B_1 \cup \{6,8\}$ ($B_2 \cup \{7,9\}$) is a biconnected component, (b) joint graph $G^{\text{J}}$; noisy vertices associated with the non-trivial blocks containing $B_1$ and $B_2$, and some other vertices are not numbered to reduce the clutter, and (c) the articulated set tree $\mathcal{T}_{\text{AST}}(G)$.

$\{17, 18, 19\}$ are associated to vertices in the articulated set tree representation, and $4, 6, 7, 14$, and $17$ are examples of articulation points.

**Definition 3.1** (Articulated Set Tree). *For an undirected graph $G = (V, E)$, the **articulated set tree** $\mathcal{T}_{\text{AST}}(G)$ is a tuple $(\mathcal{P}, \mathcal{E}, \mathcal{A})$ where (a) the set $\mathcal{P} = \{B : B \in \mathcal{B}^{\text{NT}}\} \cup \{\{v\} : v \in V \setminus \cup_{B \in \mathcal{B}^{\text{NT}}} B\}$; (b) an edge $\{P, P'\} \in \mathcal{E}$ if and only if (i) vertices $v, v' \in V$ are such that $v \in P, v' \in P'$, and $\{v, v'\} \in E$ or (ii) there exists a vertex $v \in V$ such that $v \in P \cap P'$, and (c) the articulation function $\mathcal{A} : \mathcal{E} \to V \times V$ returns the articulation points of each edge.*

Notice that the articulated set tree (AST), as the name suggests, is indeed a tree. Otherwise, by definition, the set of non-trivial blocks $\mathcal{B}^{\text{NT}}$ would be incorrect (we show this formally in Lemma B.1 in the Appendix). Readers familiar with graph theory may have observed that the AST representation is quite similar to the block-cut tree representation (see e.g., Harary, 1971; Biggs et al., 1986), but unlike a block-cut tree, the subgraph associated with any non-trivial block does not matter in the articulated set tree.

We will now define the equivalence class of graphs up to which robust recovery is possible. Let $L(G)$ denote the set of all leaves in $G$ (i.e., all vertices of degree one). A subset $R \subset L(G)$ is said to be *remote* if no two elements of $R$ share a common neighbor. Let $\mathcal{R}$ be the set of all remote subsets of $L(G)$. For each $R \in \mathcal{R}$, define a graph $G_R$ on $V$ by exchanging each vertex in $R$ with its (unique) neighbor.

**Definition 3.2** (Equivalence Relation, $\sim$). *Two graphs $G, H$ are said to be equivalent if and only if $\exists R \in \mathcal{R}$ such that $\mathcal{T}_{\text{AST}}(G_R) = \mathcal{T}_{\text{AST}}(H)$. Symbolically, we write as $H \sim G$.*

We let $[G]$ denote the equivalence class of $G$ with respect to $\sim$. It is not hard to verify that Definition 3.2 is a valid equivalence relation. Furthermore, it can be readily checked that this notion of equivalence subsumes the ones defined for trees in Katiyar et al. (2019); Casanellas et al. (2024). Fig. 2 illustrates three graphs from the same equivalence class. Notice that $G_1$ can be constructed from $G$ by: (i) exchanging the labels between the leaf vertices $\{13, 17, 15\}$ with their corresponding neighbors $\{10, 21, 14\}$; (ii) adding an edge $\{2, 4\}$ inside a non-trivial block. Similarly, $G_2$ can be constructed from $G$ by: (i) exchanging the labels between the leaf vertices $\{11, 20\}$ with their corresponding neighbors $\{10, 21\}$; (ii) removing an edge $\{1, 4\}$ from a non-trivial block[3]. Therefore, for any two graphs in the equivalence class, the non-cut vertices of any non-trivial block remain unchanged, whereas, the edges in the non-trivial block can be arbitrarily changed; the labels of the leaves can be swapped with their neighbor. In the following we will show that our identifiability result also complements the equivalence class. Finally, notice that with the similar operation described above, given a $\mathcal{T}_{\text{AST}}$ which equals to $\mathcal{T}_{\text{AST}}(G)$, one can recover all the graphs in $[G]$.

We now state our unidentifiability result, which establishes that the true covariance matrix of any graph in the equivalence class $[G]$, under the noise model of Subsection 3.1, will result in the same observed covariance matrix $\Sigma^o$.

**Theorem 3.1** (Undidentifiability). *Fix a covariance matrix $\Sigma^*$ whose conditional independence structure is given by the graph $G$. Suppose we are given a noisy covariance matrix $\Sigma^o = \Sigma^* + D$ where $D =*

---

[3]Notice that the sets $\{13, 17, 15\}$ and $\{11, 20\}$ are remote according to Definition 3.2.

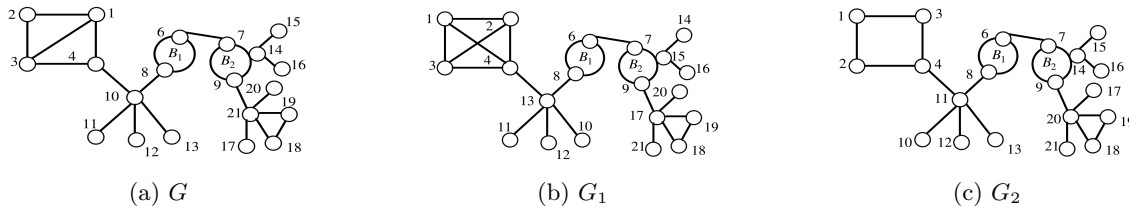

| (a) $G$ | (b) $G_1$ | (c) $G_2$ |

Figure 2: An illustration of three graphs from the same equivalence class of $G$ in Fig. 1a. For all three graphs, $B_1 \cup \{6, 8\}$ ($B_2 \cup \{7, 9\}$) can have any induced subgraph as long as subgraphs on $B_1 \cup \{6, 8\}$ ($B_2 \cup \{7, 9\}$) is a biconnected component.

$diag(D_{11}, \ldots, D_{pp}) \geq 0$. *Then, for any $H \in [G]$ where $H \neq G$, there exists matrices $\Sigma_H, D_H$ such that $\Sigma^o = \Sigma_H + D_H$, $D_H$ is a diagonal matrix with non-negative entries, and the sparsity pattern of $(\Sigma_H)^{-1}$ is described by $H$.*

This theorem is proved in Section D. Notice that this undentifiability result shows it is impossible to uniquely recover $G$ as there are other confounding graphs whose noisy observations would be indistinguishable from those of $G$. However, this theorem does not rule out the possibility of recovering the equivalence class to which $G$ belongs since all the confounding examples are confined to $[G]$. In Section 3.3, we devise an algorithm that precisely does this. Before we conclude this section, we introduce the notion of partial structure recovery and discuss how recovering the equivalence class still reveals useful information about the true graph.

**Partial Structure Recovery of $G$.** Given the noise model described in Subsection 3.1 we know that any graph is only identifiable upto the equivalence relation in Definition 3.2. This is not only the best one can do (by Theorem 3.1), but also preserves useful *partial* structure of the graph. In particular, such a partial structure recovery is able to identify the (non-cut) constituents of the non-trivial blocks and the set of leaf vertices (and neighbors thereof). As the following examples illustrate, we conclude this subsection by arguing that **even such partially recovered graphs** are instrumental in several application domains.

1. **Electrical distribution networks** usually have radial (or globally tree-like) network structures. Recently, several parts of such networks have become increasingly locally interconnected due to the adoption of technologies like roof-top solar panels and battery power storage that enable more flexible power flow. Nonetheless, practitioners critically rely on (learning) the global structure for most operations and maintenance tasks, such as state estimation, power flow, and cybersecurity.

2. **Neuronal networks.** Network models are commonly used to describe structural and functional connectivity in the brain Sporns (2018). An important property of networks that model the brain is their modular structure; modules or communities correspond to clusters of nodes that are densely connected (and are presumably functionally related). Such modular structure has long been regarded as a hallmark of many complex systems; see Simon (2012) for more details. Module detection, that is, understanding which nodes belong to which modules can yield important insights into how networks function and also uncover a network's latent community structure.

### 3.3 The Robust Model Selection Algorithm

We now present an algorithm that can recover the partial structure of a graph $G$ for which only noisy samples are available based on the setup from Subsection 3.1. Before we describe our algorithm, we introduce a few more concepts that will play a key role. We start with a well-known fact about the factorization of pairwise correlations for a faithful[4] Gaussian graphical model. First, recall that for two random variables $X_i$ and $X_j$, the *correlation coefficient* is defined as $\rho_{ij} \triangleq \Sigma_{ij}/\sqrt{\Sigma_{ii}\Sigma_{jj}}$.

---

[4]The global Markov property for GGMs ensures that graph separation implies conditional independence. The reverse implication need not to hold. However, for a faithful GGM, the reverse implication does hold.

**Fact 1** (see e.g., Soh & Tatikonda (2014)). *For a faithful Gaussian graphical model, $X_i \perp\!\!\!\perp X_k | X_j$ if and only if $\rho_{ik} = \rho_{ij} \times \rho_{jk}$.*

We now define information distances, which play a vital role in designing our algorithm.

**Definition 3.3** (Information distances). *For $(X_1, \ldots, X_p) \sim \mathcal{N}(\mathbf{0}, \Sigma)$, the information distance between $X_i$ and $X_j$ is defined by $d_{ij} \triangleq -\log|\rho_{ij}| \geq 0$, where $\rho_{ij}$ is the pairwise correlation coefficient between $X_i$ and $X_j$.*

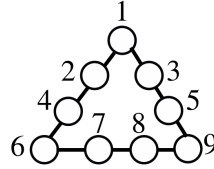

Figure 3: Graph with multiple minimal mutual separators.

For tree-structured Gaussian graphical models, the strength of the correlation dictates the information (or graphical) distance between vertices $i$ and $j$. Higher the strength, the smaller is the distance, and vice versa. In fact, the information distance defined this way is an additive metric on the vertices of the tree. Although the graphs we consider are not necessarily trees, we still refer to this quantity as a distance throughout the paper for convenience. We now define the notion of ancestors for a triplet of vertices using the notion of minimal mutual separator.

**Definition 3.4** (Minimal mutual separators, Star triplets, Ancestors). *Fix a triplet of vertices $U \in \binom{V}{3}$. A vertex set $S \subseteq V$ is called a* mutual separator *of $U$ if $S$ separates each pair $i, j \in U$; that is, every path $\pi \in \mathcal{P}_{ij}$ contains at least one element of $S$. The set $S$ is called a* minimal mutual separator *of the triplet $U$ if no proper subset of $S$ is a mutual separator of $U$. We let $S_{\min}(U)$ denote the set of all minimal mutual separators of $U$. $U$ is said to be a* star triplet *if $|S_{\min}(U)| = 1$ and the separator in $S_{\min}(U)$ is a singleton. The unique vertex that mutually separates $U$ is called the* ancestor *of $U$.*

For instance, for the graph in Fig. 3, the set $\{2, 4, 7, 8, 3, 5\}$ is a mutual separator for the triple $\{1, 6, 9\}$. Further notice that minimal mutual separator set may not be unique for a triplet: here, the sets $\{2, 3, 7\}$ and $\{4, 5, 8\}$ are minimal mutual separators of the triple $\{1, 6, 9\}$. In Fig. 1a, for the triplet $\{1, 11, 12\}$, minimal mutual separator $S_{\min}(\{1, 11, 12\}) = \{10\}$. Further, notice that an ancestor of $U$ can be one of the elements of $U$. In Fig. 1a, the vertex $\{10\}$ is the ancestor of triplet $\{4, 10, 11\} \triangleq U$. Notice that for triplet $U$, the distance between 10 and the ancestor is zero.

For a star triplet $\{i, j, k\}$ with ancestor $r$, it is clear that the following holds true based on the relationship between graph separation and conditional independence: $X_i \perp\!\!\!\perp X_j \mid X_r$, $X_i \perp\!\!\!\perp X_k \mid X_r$, and $X_j \perp\!\!\!\perp X_k \mid X_r$. As a consequence, from Fact 1 and Definition 3.3, the pairwise distances $d_{ij}, d_{ik}$, and $d_{jk}$ satisfy the following equations: $d_{ij} = d_{ir} + d_{rj}$, $d_{ik} = d_{ir} + d_{rk}$, and $d_{jk} = d_{jr} + d_{rk}$. Some straightforward algebra results in the following identities that allows us to compute the distance between each vertex in $\{i, j, k\}$ and the ancestor vertex $r$. In particular, for any ordering $\{x, y, z\}$ of the set $\{i, j, k\}$ notice that the following is true:

$$d_{xr} = 0.5 \times (d_{xy} + d_{xz} - d_{yz}) \tag{1}$$

For a triplet $U = \{i, j, k\}$, we will let $d_i^U \triangleq \frac{1}{2}(d_{ij} + d_{ik} - d_{jk})$. If $U$ is a star triplet, then $d_i^U$ reveals the distance between $i$ and the ancestor of $U$. However, we do not restrict this definition to star triplets alone. When $U$ is not a star triplet, $d_i^U$ is some arbitrary (operationally non-significant) number; in fact, for non-star triplets this quantity may even be negative. Notice that all vertex triplets in a tree are star triplets. Hence, for a tree-structured graphical model, we can choose any arbitrary triplet and if we can find a vertex $r$ for which $d_i^U = d_{ir}, i \in U$, then we can identify the ancestor of $U$. If such a vertex does not exist, we can deduce the existence of a latent ancestor. Therefore, iterating through all possible triplets, one can recover the true (latent) tree structure underlying the observed variables. In fact, several algorithms in the literature use similar techniques to learn trees (see e.g., Saitou & Nei (1987); Krishnamurthy & Singh (2012); Choi et al. (2011); Dasarathy et al. (2014).

### 3.4 The NoMAD Algorithm

In this section, we introduce our algorithm NoMAD, for Noisy Model selection based on Ancestor Discovery, for robust model selection. We describe NoMAD in the population setting (i.e., in the infinite sample limit) for clarity of presentation; the modifications required for the finite sample setting are discussed in Section 5.

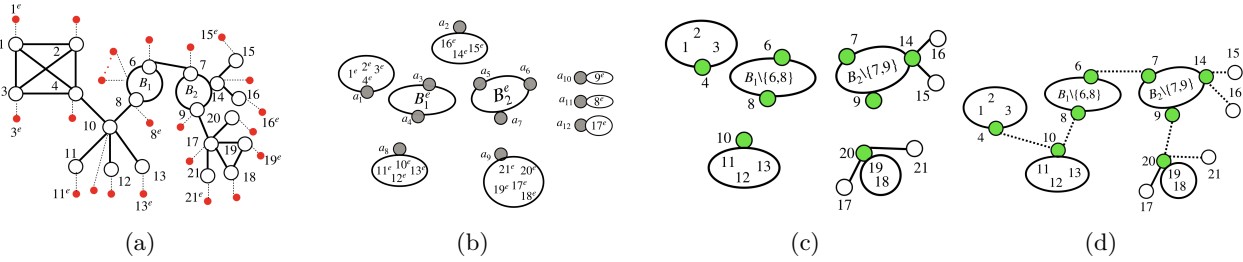

Figure 4: (a) The joint graph $G^{\mathrm{J}}$, (b) the leaf clusters and internal clusters of $G^{\mathrm{J}}$; $B_1^e$ ($B_2^e$) denote the set of noisy vertices associated with the vertices in $B_1$ ($B_2$); Also, recall that clusters are a set of vertices; grey vertices are identified but unlabeled articulation points associated with the clusters, (c) non-trivial blocks and trivial blocks along with the identified and labeled articulation points, and (d) the edges between different articulation points.

In the population setting, NoMAD takes as input the (exact) pairwise information distances $d_{ij}$, for all $i, j$ in the observed vertex set $V^{\mathrm{o}}$, and returns an articulated set tree $\mathcal{T}_{\mathrm{op}} \triangleq (\mathcal{P}_{\mathrm{op}}, A_{\mathrm{op}}, E_{\mathrm{op}})$ (see Definition 3.1). A high level overview of the algorithm is given in Algorithm 1. Its operation may be divided into two main steps: (a) learning $\mathcal{P}_{\mathrm{op}}$ and $A_{\mathrm{op}}$; and (b) learning $E_{\mathrm{op}}$. These steps are summarized in the following. A formal algorithmic listing and a full description can be found in the Appendix.

**(a) Learning $\mathcal{P}_{\mathrm{op}}$ and $A_{\mathrm{op}}$.** Inspired by the aforementioned ancestor based tree reconstruction algorithms, NoMAD first identifies the ancestors[5] in $G^{\mathrm{J}}$, and learns the pairwise distances between them. Notice that finding the ancestors in $G^{\mathrm{J}}$ is challenging for the following reasons: (a) since $G^{\mathrm{J}}$ is not a tree, some vertex triplets are not star triplets (e.g., $\{1^e, 3^e, 11^e\}$ in Fig. 1b), and (b) a subset of vertices (which may include ancestors) in $G^{\mathrm{J}}$ are unobserved or latent. Hence, we can not guarantee the identification of a star triplet following the procedure for trees. NoMAD instead uses a novel procedure that compares *two triplets* of vertices to identify the ancestors; we call this the TIA (Test Identical Ancestor) test which is defined as follows:

---

**Algorithm 1** NoMAD

---

1: **Input:** Pairwise distances $\mathcal{D} = \{d_{ij}\}_{i,j \in V^{\mathrm{o}}}$.

2: **Output:** $\mathcal{T}_{\mathrm{op}} \triangleq (\mathcal{P}_{\mathrm{op}}, A_{\mathrm{op}}, E_{\mathrm{op}})$.

3: IDENTIFYANCESTORS(Subroutine 3). **Accepts:** $\mathcal{D}$; **Returns:** (I) A set $A_{\mathrm{obs}}$ ($A_{\mathrm{hid}}$ resp.) of observed (hidden resp.) ancestors, and the corresponding collection of vertex triplets $\mathfrak{V}_{\mathrm{obs}}$ ($\mathfrak{V}_{\mathrm{hid}}$ resp.), and (II) The set of pairwise distances $\{d_{ij}\}$ for each pair $i, j \in V^{\mathrm{o}} \cup A_{\mathrm{hid}}$

4: LEARNCLUSTERS (Subroutine 4). **Accepts:** $A_{\mathrm{obs}}, A_{\mathrm{hid}}$, and $\mathcal{D}$; **Returns:** A collection of (I) leaf clusters $\mathcal{L}$, and (II) internal clusters $\mathcal{I}$.

5: VERTEXSET-AST(Subroutine 6).**Accepts:** $\mathcal{L}$ and $\mathcal{I}$; **Returns:** (I) the vertex set $\mathcal{P}_{\mathrm{op}}$ (II) the articulation points $A_{\mathrm{op}}$, and (III) a *subset* of the edge set $E_{\mathrm{op}}$.

6: EDGESET-AST(Subroutine 8). **Accepts:** $\mathcal{P}_{\mathrm{op}}$ and $A_{\mathrm{op}}$; **Returns:** $E_{\mathrm{op}}$.

---

**Definition 3.5** (TIA Test). *The Test Identical Ancestor (TIA) test accepts a triplet pair $U, W \in \binom{V^{\mathrm{o}}}{3}$, and returns TRUE if and only if for all $x \in U$, there exists at least one pair $y, z \in W$ such that $d_x^U + d_y^W = d_{xy}$ and $d_x^U + d_z^W = d_{xz}$.*

In words, in order for a pair of triplets $U, W$ to share an ancestor in $G^{\mathrm{J}}$, each vertex in one triplet (say, $U$) needs to be separated from at least a pair in $W$ by the (shared) ancestor in $G^{\mathrm{J}}$. We now describe the fist step of the NoMAD in three following sub-steps:

Identifying the ancestors in $G^{\mathrm{J}}$. In the first sub-step, NoMAD (a) uses the TIA test to create the set $\mathfrak{V} = \{\mathcal{V} \subset \binom{V}{3} : \text{each } U \in \mathcal{V} \text{ has the same ancestor }\}$, (b) it then assigns each triplet collection $\mathcal{V} \in \mathfrak{V}$ to

---

[5]We say that a vertex $r$ is an ancestor if there is a triple $U$ such that $r$ is an ancestor of $U$.

either $\mathfrak{V}_{\text{obs}}$ or $\mathfrak{V}_{\text{hid}}$; the former is the collection of vertex triples whose ancestor is observed and the latter has ancestors that are hidden, and (c) identifies the observed ancestors and enrolls them into a set of observed ancestors $A_{\text{obs}}$. Furthermore, for each collection $\mathcal{V}_i \in \mathfrak{V}_{\text{hid}}$, NoMAD introduces a hidden vertex, and enrolls it in $A_{\text{hid}}$ such that, $|A_{\text{hid}}|$ equals to the number of hidden ancestors in $G^{\text{J}}$. For example, consider the joint graph $G^{\text{J}}$ in Fig. 4a. In $G^{\text{J}}$, the vertex $\{4\}$ is the observed ancestor of the pair $\{1^e, 4^e, 10\}$ and $\{3^e, 4^e, 8^e\}$, and the vertex $\{8\}$ is the hidden ancestor of the pair $\{3^e, 8^e, 9^e\}$, $\{1^e, 8^e, 7^e\}$. Complete pseudocode for this step appears in Subroutine 3 in Appendix.

**Extending the distance set** $\{d_{ij}\}_{i,j \in V^o}$. In the next sub-step, using pairwise distances $\{d_{ij}\}_{i,j \in V^o}$, and $A_{\text{hid}}$, NoMAD learns the following distances: (a) $d_{ij}$ for all $i, j \in A_{\text{hid}}$, and (b) $d_{ij}$ for all $i, j \in A_{\text{hid}} \cup V^o$. For learning (a), notice from the last sub-step that each $a_i \in A_{\text{hid}}$ is assigned to a collection of triplets $\mathcal{V}_i \in \mathfrak{V}_{\text{hid}}$. In order to compute the distance between two hidden ancestors (say $a_p, a_q \in A_{\text{hid}}$), the NoMAD chooses two triplets $V_p \in \mathcal{V}_p$ and $V_q \in \mathcal{V}_q$, and computes the set $\Delta_{pq}$ as follows: $\Delta_{pq} = \left\{ d_{xy} - (d_x^{V_p} + d_y^{V_q}) : x \in V_p, y \in V_q \right\}$. Then, the most frequent element in $\Delta_{pq}$, i.e., $\text{mode}(\Delta_{pq})$, is declared as $d_{pq}$. We show in Appendix that NoMAD not only correctly learns the distance set $\{d_{ij}\}_{i,j \in A_{\text{hid}}}$, but also learns $d_{ij}$ for any $i \in A_{\text{hid}}$ and $j \in V^o$. A pseudocode for this step appears in Subroutine 3 in Appendix.

**Learning** $\mathcal{P}_{\text{op}}$ **and** $A_{\text{op}}$. In the final sub-step, NoMAD learns the clusters of vertices in $V^o \setminus A_{\text{obs}}$ using the separation test in Fact 1 which eventually lead to finding $\mathcal{P}_{\text{op}}$ and $A_{\text{op}}$. Specifically, NoMAD learns (a) all the leaf clusters, each of which is a set of vertices that are separated from the rest of the graph by a single ancestor, and (b) all the internal clusters, each of which is a set of vertices that are separated from the rest of the graph by multiple ancestors. For example, in Fig. 4b, the set $\{17^e, 18^e, 19^e, 20^e, 21^e\}$ is a leaf cluster— separated from the rest of the graph by the (hidden) ancestor $a_9$. The set $B_2^e$ is an internal cluster— separated from the rest of the graph by the set of ancestors $\{a_5, a_6, a_7\}$. Next, NoMAD uses the clusters to learn $\mathcal{P}_{\text{op}}$ and $A_{\text{op}}$ for $\mathcal{T}_{\text{op}}$ by applying the TIA test on each cluster to identify the non-cut vertices and potential cut vertices in it. For example, for the leaf cluster $\{17^e, 18^e, 19^e, 20^e, 21^e\}$, 18 and 19 are non-cut vertices, and a vertex from $17, 20,$ and $21$ may be declared as a cut-vertex arbitrarily (see Fig. 4c). A pseudocode for this step appears as Subroutine 4 and Subroutine 6 in Appendix.

**(b) Learning** $E_{\text{op}}$. In this step, NoMAD learns the edge set $E_{\text{op}}$. Notice from Definition 3.1 that any two elements of $\mathcal{P}_{\text{op}}$ are connected with each other through their respective articulation points. Hence, in order to learn $E_{\text{op}}$, NoMAD needs to learn the neighborhood of each articulation point in $G$. To this end, NoMAD first learns this neighborhood for each articulation point in $G$ using Fact 1. Then, in the next step, NoMAD creates an edge between two elements of $\mathcal{P}_{\text{op}}$ if the articulation points from each element are neighbors in $G$ (dotted lines in Fig. 4d). A pseudocode for this step appears as Subroutine 8 in Appendix.

## 4 Performance Analysis of NoMAD in the Population Setting

In this section, we show the correctness of NoMAD in returning the equivalence class of a graph $G$ while having access only to the noisy samples according to the problem setup in Section 3.1. We now make an assumption that will be crucial to show the correctness of NoMAD. This is similar to the faithfulness assumption common in the graphical modeling literature Choi et al. (2011); Kalisch & Bühlman (2007); Uhler et al. (2013), and like the latter, it rules out "spurious cancellations". To that end, let $\mathcal{V}_{\text{star}} \subseteq \binom{V}{3}$ be the set of all star triplets in $G$ (see Definition 3.4). Let $\mathcal{V}_{\text{sep}} \subseteq \binom{V}{3}$ be the set of triplets $V$ such that one of the vertices in $V$ separates the other two vertices.

**Assumption 4.1** (Ancestor faithfulness). *Let* $U, W \in \binom{V}{3} \setminus \mathcal{V}_{\text{star}} \cup \mathcal{V}_{\text{sep}}$. *Then, (i) there are no vertices* $x \in U$ *and* $a \in W$ *that satisfy* $d_x^U + d_a^W = d_{xa}$, *and (ii) there does not exist any vertex* $r \in V$ *and* $x \in U$ *for which the distance* $d_{xr}$ *satisfies relation in equation 1.*

Notice that Assumption 4.1 is only violated when there are explicit constraints on the corresponding covariance values which, like the faithfulness assumption, only happens on a set of measure zero. We next state the main result of our paper in the population settings.

**Theorem 4.2.** *Consider a covariance matrix $\Sigma^*$ whose conditional independence structure is given by the graph $G$, and the model satisfies Assumption 4.1. Suppose that according to the problem setup in Section 3.1, we are given pairwise distance $d_{ij}$ of a vertex pair $(i, j)$ in the observed vertex set $V^o$, that is, $d_{ij} \triangleq -\log|\rho_{ij}|$ where $\rho_{ij} \triangleq \Sigma_{ij}^o / \sqrt{\Sigma_{ii}^o \Sigma_{jj}^o}$. Then, given the pairwise distance set $\{d_{ij}\}_{i,j \in V^o}$ as inputs, NoMAD outputs the equivalence class $[G]$.*

**Proof Outline.** In order to show that NoMAD correctly learns the equivalence class, it suffices to show that it can correctly deduce the articulated set tree $\mathcal{T}_{\mathrm{op}}$. Given this, and the equivalence relation from Definition 3.2, the entire equivalence class can be readily generated. Our strategy will be to show that NoMAD learns $\mathcal{T}_{\mathrm{op}}$ correctly by showing that it learns (a) the vertex set $\mathcal{P}_{\mathrm{op}}$, (b) the articulation points $A_{\mathrm{op}}$, and (c) the edge set $E_{\mathrm{op}}$ correctly. We will now establish (a) and (b).

- From the description of the algorithm in Section 3.3, it is clear that NoMAD succeeds in finding the ancestors, which is the first step, provided the TIA tests succeed. Indeed, in the first stage of our proof, we establish Lemma B.7 in Appendix which shows that the TIA test passes with two triplets if and only if they share a common ancestor in $G^{\mathrm{J}}$.

- Next, we show in Lemma B.9 and Claim 3 in Appendix that NoMAD correctly learns the distances $d_{ij}$ for all vertices $i, j$ that either in the set of observed vertices $V^o$ or in the set of hidden ancestors $A_{\mathrm{hid}}$. Proposition B.13 establishes the correctness of NoMAD in learning the leaf clusters and internal clusters, and in learning $\mathcal{P}_{\mathrm{op}}$ and $A_{\mathrm{op}}$. The proof correctness of this step crucially depends on identifying the non-cut vertices in $G$ from different clusters which is proved in Lemma B.12.

Finally, we outline the correctness of NoMAD in learning the edge set $E_{\mathrm{op}}$. Recall that NoMAD learns the neighbor articulation points of each articulation point. Proposition B.14 in Appendix shows that NoMAD correctly achieves this task. Using the neighbors of different articulation points, NoMAD correctly learns the edges between different elements in $\mathcal{P}_{\mathrm{op}}$.

## 5 Performance Analysis of NoMAD in Finite Sample Setting

In describing NoMAD (cf. Section 3.4) and in the analysis in the population setting (cf. Section 4), we temporarily assumed that we have access to the actual distances $d_{ij}$ for the sake of exposition. However, in practice, these distances need to be estimated from samples $\{\mathbf{Y}_1, \ldots, \mathbf{Y}_n\}$. In what follows, we show that NoMAD, with high probability, correctly outputs the equivalence class $[G]$ even if we replace $d_{ij}$ with the estimate $\widehat{d}_{ij} = -\log\left|\widehat{\Sigma^o}_{ij} / \sqrt{\widehat{\Sigma^o}_{ii}\widehat{\Sigma^o}_{jj}}\right|$, where $\widehat{\Sigma^o}_{ij}$ is the $(i, j)$-th element in $\frac{1}{n}\sum_{i=1}^{N}\mathbf{Y}_i\mathbf{Y}_i^T$. We recall from Section 3.4 that the subroutines in NoMAD depend on the TIA test, which relies on the distances $d_{ij}$. Thus, we establish the correctness of NoMAD in the finite sample setting by showing that the empirical TIA (TIA with estimated distances) correctly identifies the ancestors in $G^J$ with high probability. We begin with the following assumptions.

**Assumption 5.2.** *[$\gamma$-Strong Faithfulness Assumption] For any vertex triplet $i, j, k \in \binom{V^o}{3}$, if $i \not\perp\!\!\!\perp j|k$, then $|d_{ij} - d_{ik} - d_{jk}| > \gamma$.*

$\gamma$-Strong Faithfulness assumption is a standard assumption used in the literature (Kalisch & Bühlman (2007); Uhler et al. (2013)). 0-Strong-Faithfulness is just the usual Faithfulness assumption discussed in Section 3.3. This motivates our next assumption which strengthens our requirement on "spurious cancellations" involving ancestors.

**Assumption 5.3** (Strong Ancestor consistency). *For any triplet pair $U, W \in \binom{V}{3} \setminus \mathcal{V}_{\mathrm{star}} \cup \mathcal{V}_{\mathrm{sep}}$ and any vertex pair $(x, a) \in U \times W$, there exists a constant $\zeta > 0$, such that $\left|d_x^U + d_a^W - d_{xa}\right| > \zeta$.*

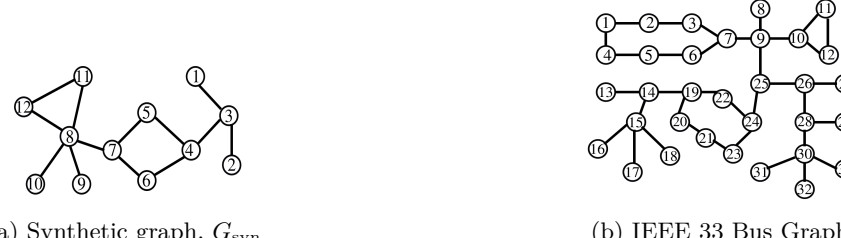

(a) Synthetic graph, $G_{\text{syn}}$         (b) IEEE 33 Bus Graph

Figure 5: Synthetic graph and IEEE-33 Bus system considered for our simulation

Assumption 5.3 is in direct analogy with Assumption 5.2. As we show in Lemma B.7, for any pair $U, W \in \binom{V}{3} \setminus \mathcal{V}_{\text{star}} \cup \mathcal{V}_{\text{sep}}$ (i.e., any pair that would fail the TIA test), there exists at least one triplet $\{x, a, b\}$ where $x \in U$ and $a, b \in W$ such that $d_{xa} - d_x^U - d_a^W \neq 0$ and $d_{xb} - d_x^U - d_b^W \neq 0$. This observation motivates us to replace the exact equality testing in the TIA test in Definition 3.5 with the following hypothesis test against zero: $\max\left\{\left|\widehat{d}_{xa} - \widehat{d}_x^U - \widehat{d}_a^W\right|, \left|\widehat{d}_{xb} - \widehat{d}_x^U - \widehat{d}_b^W\right|\right\} \leq \xi$. We set $\xi < \frac{\zeta}{2}$.

Furthermore, in order to learn the distance between two hidden ancestors in $A_{\text{hid}}$, the mode test introduced in Subsection 3.4 needs to be replaced with a finite sample version; we call this the $\epsilon_d$-mode test and this is formalized in Appendix C. Finally, for any triplet $(i, j, k) \in \binom{V^o}{3}$, in order to check whether $i \perp\!\!\!\perp j | k$, the test in Fact 1 needs to be replaced as follows: $|\widehat{d}_{ij} - \widehat{d}_{ik} - \widehat{d}_{jk}| < \frac{\epsilon_d}{6}$. We now introduce two new notations to state our main result. Let $\rho_{\min}(p) = \min_{i,j \in \binom{p}{2}} |\rho_{ij}|$ and $\kappa(p) = \log((16 + (\rho_{\min}(p))^2 \epsilon_d^2)/(16 - (\rho_{\min}(p))^2 \epsilon_d^2))$, where $\epsilon_d = \min(\frac{\xi}{14}, \gamma)$, where $\gamma$ is from Assumption 5.2.

**Theorem 5.3.** *Suppose the underlying graph $G$ of a faithful GGM satisfies Assumptions 5.2-5.3. Fix any $\tau \in (0, 1]$. Then, there exists a constant $C > 0$ such that if the number of samples $n$ satisfies $n > C\left(\frac{1}{\kappa(p)}\right) \max\left(\log\left(\frac{p^2}{\tau}\right), \log\left(\frac{1}{\kappa(p)}\right)\right)$, then with probability at least $1 - \tau$, NoMAD accepting $\widehat{d}_{ij}$ outputs the equivalence class $[G]$.*

**Remark 5.1.** *Theorem 5.3 indicates that the sample complexity of the NoMAD is dependent on the absolute minimum and maximum pairwise correlations $\rho_{\min}$ and $\rho_{\max}$, the number of vertices $p$, and the magnitude of the quantity from Assumption 5.3. Specifically, in regimes of interest, we see that the sample complexity scales as a logarithm in the number of vertices $p$ and inversely in $\rho_{\min}^2(p)$ and $\epsilon_d^2$, thus allowing for robust model selection in the high-dimensional regime. Further note that our sample requirement can have an exponential dependence on $p$ based on the degree of $G$. However, we expect that this dependency can be improved; see Section 7 for more on this avenue for future work.*

## 6 Experiments

We perform experiments on both a synthetic graph and an IEEE 33 bus system, which is a graphical representation of established IEEE-33 bus benchmark distribution system (Baran & Wu, 1989), to assess the validity of our theoretical results and to demonstrate the performance of NoMAD. In particular, our experiments demonstrate how an unmodified graphical modeling algorithm (in this case GLASSO (Friedman et al. (2008)) compares to NoMAD. The synthetic graph $G_{\text{syn}}$ we consider and graph associated with the IEEE-33 bus system are given in Fig. 5a and Fig. 5b, respectively. As a first step, we need some new performance metrics to make a fair comparison in light of the unidentifiability of the underlying graph structure from noisy data. In the following we will introduce some sets, and show that if an algorithm can recover all these sets, then the output graph (from that algorithm) is in equivalence class.

1. **Families.** For a vertex $i$, define its *family* $F_i$ as $\{v : \deg(v) = 1 \text{ and } \{v, i\} \in E(G)\} \cup \{i\}$. Notice that the subgraph associated with each family is a tree. For synthetic graph $G_{\text{syn}}$, the sets $\{1, 2, 3\}$, $\{8, 9, 10\}$ are some of the families in $G_{\text{syn}}$; For IEEE 33 Bus Graph $\{15, 16, 17, 18\}$, $\{30, 31, 32, 33\}$ are some of the families. Let $\mathcal{F} = \bigcup_{i \in V} F_i$.

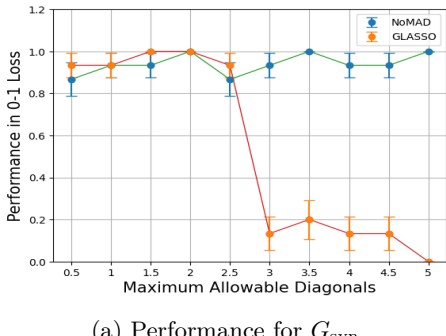

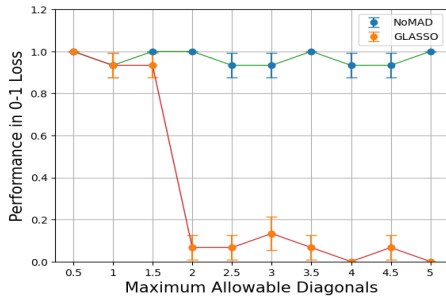

| (a) Performance for $G_{\text{syn}}$ | (b) Performance for IEEE-33 Bus. |
|---|---|

Figure 6: Performance of GLASSO and NoMAD for various degree of noises. Up to the values of 2.5 and 1.5 dollars for $G_{\text{syn}}$ and IEEE-33 Bus, respectively, GLASSO successfully reconstructed an equivalent graph. To investigate this influence, we conducted the experiment with a fixed sample size across 15 trials. In contrast, NoMAD consistently demonstrated its ability to recover an equivalent graph under various noise influences.

2. **Non-Cut Vertices.** For graph $G$, let $B_{\text{non-cut}}$ be the set of all non-cut vertices in a non-trivial block $B$. Define $\mathcal{B}_{\text{non-cut}} \triangleq \bigcup_{B \in \mathcal{B}^{\text{NT}}} B_{\text{non-cut}}$, where $\mathcal{B}^{\text{NT}}$ is the set of all non-trivial blocks. Recall from Section 3 that in all the equivalent graphs the set of non-cut vertices remain unchanged.

3. **Global Edges.** Let $K$ be the set of cut vertices who do not share an edge with a leaf vertex in $G$. For any vertex $k \in K$, let a family $F_k \in \mathcal{F}$ be such that there exists a vertex $f \in F_k$ such that $\{k, f\} \in E(G)$. Now, we will note two following condition for any neighbor $i \in N(k)$: (a) if $i \in K$, then $\{i, k\} \in E(G)$, and (b) otherwise, there exists a vertex $j \in F_k$ such that $\{j, k\} \in E(G)$. If condition (a) and condition (b) are met, global edge associated with $i$ are recovered correctly .

In Lemma B.15, we demonstrate that if an algorithm learns these sets correctly, and the conditions associated with the global edges are met, then an equivalent graph can be recovered.

**Experimental Setup**. We now describe our experimental setup. We generated precision matrices associated with $G_{\text{syn}}$, and for IEEE-33 bus system, we added some loops. We then inverted the corresponding precision matrices to obtain the respective covariance matrices in the population setting. Next, we added a diagonal matrix of random positive values to the population covariance matrices at each entry to generate the corresponding noisy covariance matrices.

We will now compare the performance of 0-1 loss of NoMAD to GLASSO under the influence of noise. In order to compare, our goal is to compare NoMAD with the *best* GLASSO. Our protocol for selecting the *best* GLASSO is as follows: for a fixed maximum allowable diagonals, we selected the regularization parameter for which the output graph given the noise covariance matrix to GLASSO is in equivalence class. Then, for that fixed regularization parameter we report the performance of GLASSO for varying numbers of (increased) maximum allowable diagonals. Notice that the maximum allowed values of the diagonal elements of the matrix $D$ (which is being added to the generated covariance matrix) contain information about the potential influence of the noise in the setup. In order to study this influence we ran the experiment for a fixed sample size over 15 trials. In Fig. 6, we observe that up to the value of 2.5 and 1.5 for $G_{\text{syn}}$ and IEEE-33 Bus, respectively, GLASSO was able to recover an equivalent graph. On the contrary, NoMAD was able to show a consistent performance in recovering an equivalent graph with various influence of noises.

## 7 Conclusion and Future Directions

**Conclusion.** In this paper, we consider model selection of Gaussian graphical models when the observations are corrupted by independent but non-identically distributed noise with unknown statistics. We first show that this ambiguity is unavoidable. Finally, we devise a novel algorithm, referred to as NoMAD, which learns

structure up to a small equivalence class.

**Future Directions.** This paper opens up several exciting avenues for future research. First, our novel ancestor testing method can be used to identify the ancestors for other graphical models (beyond Gaussians) where information distance satisfies the factorization property in Fact 1; e.g., Discrete graphical models, where the random vector $\mathbf{X}$ takes values in the product space $\mathcal{X}^p$, where $|\mathcal{X}| = k$. For any $i, j \in [p]$, let $\Upsilon_{ij} \in \mathbb{R}^{k \times k}$ denote the tabular representation of the marginal distribution of the pair $(X_i, X_j)$ and $\Upsilon_{ii}$ denote a diagonal matrix with the marginal distribution of $X_i$ on the diagonal. Then, it is known that the following quantity: $d_{ij} = \frac{\det(v_{ij})}{\sqrt{\det(v_{ii} v_{jj})}}$, can be taken to be the information distance. We refer the reader to Semple & Steel (2003) for more on this, including a proof of an equivalent version of Fact 1. Note that the Ising model is a special case, and our results naturally extend to them. Second, it is well known that $\rho_{\min}$ could scale exponentially in the diameter of the graph; this could imply that the sample complexity will scale polynomially in the number of vertices $p$ even for balanced binary trees, and as bad as exponential for more unbalanced graphs. Now, notice that in Subroutine 3, NoMAD identifies all the star triplets for any ancestor in $G^{\mathsf{J}}$. Hence, this identification procedure in quite computationally expensive. As we reason this in theoretical section of the appendix that this computation is required in order to learn the pairwise distances $d_{ij}$ for each pair $(i, j)$ such that $i \in V^{\mathrm{o}}$ and $j \in A_{\mathrm{hid}}$, where $V^{\mathrm{o}}$ and $A_{\mathrm{hid}}$ is the set of observed vertices, and hidden ancestors, respectively. A promising future research work is to develop a TIA test which can obtain $\{d_{ij}\}_{i,j \in V^{\mathrm{o}} \cup A_{\mathrm{hid}}}$ without iterating all the triplets in $\binom{V^{\mathrm{o}}}{3}$. Furthermore, the Subroutine 4 can be redesigned to a computationally efficient one by learning the clusters in *divide and conquer* manner. Another promising avenue for future research work is to obtain an upper bound on the diagonal entries $D_{ii}$ for which the underlying graph $G$ is identifiable. Finally, future research can be done to understand to understand the behavior of the hyperparameters $\gamma$ and $\zeta$ using the stability selection method (see Meinshausen & Bühlmann, 2010) in finite sample settings.

## Acknowledgments

The work was supported in part by the National Science Foundation (NSF) under the grants EPCN-2246658, OAC-1934766, and CCF-2048223. It was also supported by the National Institutes of Health (NIH) under the grant 1R01GM140468-01.

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
