# Appendix

# Robust Model Selection of Gaussian Graphical Models

## A    Algorithmic Details

In the population setting, NoMAD takes as input the pairwise distances $d_{ij}$, for all $i, j$ in the observed vertex set $V^{\mathrm{O}}$, and returns an articulated set tree $\mathcal{T}_{\mathrm{op}} \triangleq (\mathcal{P}_{\mathrm{op}}, A_{\mathrm{op}}, E_{\mathrm{op}})$ (see Definition 3.1). Its operation is divided into two main steps: (a) learning $\mathcal{P}_{\mathrm{op}}$ and $A_{\mathrm{op}}$; and (b) learning $E_{\mathrm{op}}$. These steps are summarized in the following.

### A.1    Learning $\mathcal{P}_{\mathrm{op}}$ and $A_{\mathrm{op}}$ for $\mathcal{T}_{\mathrm{op}}$

In *Phase 1*, Subroutine 3 identifies the ancestors in $G^{\mathrm{J}}$ using the pairwise distances $d_{ij}$ for all $i, j \in V^{\mathrm{O}}$. In this phase, it returns a collection $\mathfrak{V}$ of vertex triplets such that each triplet collection $\mathcal{V} \in \mathfrak{V}$ contains (and only contains) all vertex triples that share an identical ancestor in $G^{\mathrm{J}}$. The key component for this is to use TIA (Test Identical Ancestor). In the *Phase 2*, Subroutine 3 enrolls each collection in $\mathfrak{V}$ to either $\mathfrak{V}_{\mathrm{obs}}$ or $\mathfrak{V}_{\mathrm{hid}}$, such that $\mathfrak{V}_{\mathrm{obs}}$ ($\mathfrak{V}_{\mathrm{hid}}$) contains the collection of vertex triplets for which their corresponding ancestors are observed (hidden resp.), and observed ancestors are enrolled in the set $A_{\mathrm{obs}}$. For identifying the observed ancestors from $\mathfrak{V}$, Subroutine 3 does the following for each collection $\mathcal{V} \in \mathfrak{V}$: it checks for a vertex triplet $T$ in $\mathcal{V}$ for which one vertex in the triplet $T$ separates the other two. In the final phase, Subroutine 3 accepts $d_{ij}$ for each pair $i, j \in V^{\mathrm{O}}$ and $\mathfrak{V}_{\mathrm{hid}}$, and learns the pairwise distance $d_{ij}$ for each $i \in V^{\mathrm{O}}$ and $j \in A_{\mathrm{hid}}$ by finding a vertex triplet $T$ in a collection $\mathcal{V}_j \in \mathfrak{V}_{\mathrm{hid}}$ such that $T$ contains $i$. Then, in the next step, Subroutine 3 learns $d_{ij}$ for each $i, j \in A_{\mathrm{hid}}$ by selecting the most frequent distance in $\Delta_{pq}$ as defined in Section 3.4.

We next present Subroutine 4 for clustering the vertices in the set $V^{\mathrm{O}} \setminus A_{\mathrm{obs}}$. It accepts $A_{\mathrm{obs}}$, $A_{\mathrm{hid}}$, and $\{d_{ij}\}_{i \in V^{\mathrm{O}}, j \in A_{\mathrm{hid}}}$, and enrolls each vertex in $V^{\mathrm{O}} \setminus A_{\mathrm{obs}}$ either in a *leaf cluster* (see Phase 1 of Subroutine 4) or in an *internal cluster* ( see Phase 2 of Subroutine 4). Now, for the collection of leaf clusters $\mathcal{L}$, each cluster $L \in \mathcal{L}$ is associated to a unique element $a \in A$ such that $L_2$ is separated from $A \setminus a$ by $a$. Each cluster $I \in \mathcal{I}$ is associated with a subset of ancestors $I_1 \subset A$, such that $I_2$ is separated from all other ancestors in $A \setminus I_1$ by $I_1$.

---

**Procedure 2** TESTIDENTICALANCESTOR (TIA)

1:  **procedure** TIA$(U, W)$
2:    **if** for all $x \in U$, $\exists$ at least a pair $y, z \in W$ such that $d_x^U + d_y^W = d_{xy}$ and $d_x^U + d_z^W = d_{xz}$ **then**
3:      **Return** TRUE.
4:    **end if**
5:    **Return** FALSE.
6:  **end procedure**

---

**Procedure 7** NONBLOCKNEIGHBORS

1:  **Input:** An ancestor vertex $u$, $\mathcal{C}_u$, $A_{\mathrm{op}}$, and the extended distance set $\mathcal{D}_{\mathrm{ext}}$.
2:  **Output:** Neighbors $\delta(u)$ of $u$ such that they do not belong to the clusters that contains $u$.
3:  **Initialize:** $\delta(u) \triangleq A_{\mathrm{op}} \setminus \bigcup_{C \in \mathcal{C}_u} C_3$.
4:  **for** each $x \in \delta(u)$ **do**
5:    **if** $\exists$ a vertex $b \in \mathcal{C}_u \setminus$ s.t. $d_{ux} = d_{ub} + d_{bx}$ **then**
6:      $\delta(u) \leftarrow \delta(u) \setminus x$
7:    **end if**
8:  **end for**
9:  **for** each $k, \ell \in \binom{\delta(u)}{2}$ **do**
10:    **if** $d_{uk} + d_{k\ell} = d_{u\ell}$ **then**
11:      $\delta(u) \leftarrow \delta(u) \setminus \ell$.
12:    **end if**
13:  **end for**

---

**Subroutine 8** Learning $E_{\mathrm{op}}$ for $\mathcal{T}_{\mathrm{op}}$

1:  **Input:** The collection of leaf clusters $\mathcal{L}$ and internal clusters $\mathcal{I}$, $\mathcal{C} \triangleq \mathcal{L} \cup \mathcal{I}$, a subset $\mathcal{E}_{\mathrm{leaf}}$ of $E_{\mathrm{op}}$.
2:  **Output:** An edge set $E_{\mathrm{op}}$ for $\mathcal{T}_{\mathrm{op}}$.
3:  **Initialize:** $E_{\mathrm{op}} \leftarrow \mathcal{E}_{\mathrm{leaf}}$.
4:  **for** each $u \in A_{\mathrm{op}}$ **do**
5:    Let $C \in \mathcal{C}$ be the cluster such that $C_3 \ni u$.
6:    Get $\delta(u)$ from NONBLOCKNEIGHBORS$(u, C, A_{\mathrm{op}})$.
7:    **for** each $P_u \in \mathcal{P}_{\mathrm{op}}$ s.t. $P_u \ni u$ **do**
8:      **for** each $v \in \delta(u)$ **do**
9:        $E_{\mathrm{op}} \leftarrow E_{\mathrm{op}} \cup (P_u, \{v\}, u, v)$
10:      **end for**
11:    **end for**
12:  **end for**
13:  **Return** The edge set $E_{\mathrm{op}}$ for $\mathcal{T}_{\mathrm{op}}$.

---

We now discuss NONCUTTEST appears in Procedure 5. The goal of NONCUTTEST is to learn (a) the non-cut vertices, and (b) potential cut vertices of a non-trivial block from a leaf cluster. NONCUTTEST accepts a set $W \subseteq V^{\mathrm{O}}$ s.t. $|W| \geq 3$, and partitions the vertex set $W$ into $C_{\mathrm{cut}}$ (the set of potential cut vertices) and $C_{\mathrm{non-cut}}$ (the set of vertices which *can not* be a cut vertex). Then we use Subroutine 6 for learning (a) vertex

---

**Subroutine 3** Identifying Ancestors and Extending the Pairwise Distance Set

---

1: **Input:** Pairwise distances $\mathcal{D} = \{d_{ij}\}_{i,j \in V^O}$, where $V^O$ is the set of observed vertices.
2: **Return:** A collection of vertex triplets $\mathfrak{V}_{\text{obs}}$ ($\mathfrak{V}_{\text{hid}}$ resp.) with observed (hidden resp.) ancestors, the set $A_{\text{obs}}$ ($A_{\text{hid}}$ resp.) of observed (hidden resp.) ancestors, the set of pairwise distances $\{d_{ij}\}$ for each pair $i,j \in V^O \cup A_{\text{hid}}$.
3: **Initialize:** $\mathfrak{V}_{\text{obs}}, \mathfrak{V}_{\text{hid}}, \widetilde{\mathcal{D}}, \mathcal{D}_{\text{hid}} \leftarrow \emptyset$, collection of vertex triplets $\mathcal{V} \triangleq \binom{V^O}{3}$, counter $n = 1$

*Phase 1 – Clustering Star Triplets*

4: **for** each $U \in \mathcal{V}$ **do**
5: $\quad \mathcal{V}_n \triangleq \{W \subset \mathcal{V} : \text{TIA}(U, W) \text{ is True}\} \cup U$
6: $\quad$ **if** $|\mathcal{V}_n| > 1$ **then** $n = n + 1$
7: $\quad$ **end if**
8: $\quad \mathfrak{V} \leftarrow \mathfrak{V} \cup \mathcal{V}_n$ $\qquad\qquad\qquad$ ▷ enrolling the collection $\mathcal{V}_n$ to $\mathfrak{V}$
9: **end for**
10: **Return:** $\mathfrak{V} = \{\mathcal{V} \subset \binom{V}{3} : \text{each } U \in \mathcal{V} \text{ has the same ancestor }\}$,

*Phase 2 – Labeling Ancestors*

11: **for** each collection $\mathcal{V} \in \mathfrak{V}$ **do**
12: $\quad$ **if** $\exists$ a triplet $V \triangleq \{u, v, w\} \in \mathcal{V}$ s.t. $d_{uv} + d_{vw} = d_{uw}$ **then**
13: $\qquad \mathfrak{V}_{\text{obs}} \leftarrow \mathfrak{V}_{\text{obs}} \cup \mathcal{V}$
14: $\qquad A_{\text{obs}} \leftarrow A_{\text{obs}} \cup v$

15: $\quad$ **else**
16: $\qquad \mathfrak{V}_{\text{hid}} \leftarrow \mathfrak{V}_{\text{hid}} \cup \mathcal{V}$
17: $\quad$ **end if**
18: **end for**
19: Set $A_{\text{hid}} \triangleq \{a_i | i \in [|\mathfrak{V}_{\text{hid}}|]\}$ ▷ introduce one vertex for each element in $\mathfrak{V}_{\text{hid}}$

*Phase 3 – Learning the pairwise distance set $\{d_{ij}\}_{i,j \in V^O \cup A_{hid}}$*

20: **for** each $\mathcal{V}_i \in \mathfrak{V}_{\text{hid}}$ **do**
21: $\quad$ **for** each $j \in V^O$ **do**
22: $\qquad$ Find a triplet $U \in \mathcal{V}_i$ s.t $U \ni j$ ▷ cf. Claim 3
23: $\qquad \widetilde{\mathcal{D}} \leftarrow \widetilde{\mathcal{D}} \cup \left( (i,j), d_j^U \right)$
24: $\quad$ **end for**
25: **end for**
26: **for** each $p \neq q \in A_{\text{hid}}$ **do**
27: $\quad$ Pick a pair of triplets $U_p \in \mathcal{V}_p, U_q \in \mathcal{V}_q$.
28: $\quad \Delta_{pq} = \left\{ d_{xy} - (d_x^{U_p} + d_y^{U_r}) : x \in U_p, y \in U_q \right\}$.
29: $\quad \mathcal{D}_{\text{hid}} \leftarrow \mathcal{D}_{\text{hid}} \cup \left( (p,q), \text{mode}(\Delta_{pq}) \right)$ ▷ most frequent element in $D_{pq}$
30: **end for**
31: **Return** $\mathfrak{V}_{\text{obs}}, \mathfrak{V}_{\text{hid}}, A_{\text{obs}}, A_{\text{hid}}, \widetilde{\mathcal{D}}$, and $\mathcal{D}_{\text{hid}}$.

---

**Subroutine 4** LEARNCLUSTERS

---

1: **Input:** $A_{\text{obs}}, A_{\text{hid}}$, and $\mathcal{D}$, and $A \triangleq A_{\text{obs}} \cup A_{\text{hid}}$.
2: **Output:** A collection of leaf clusters $\mathcal{L}$ and internal clusters $\mathcal{I}$.
3: **Initialize:** $\mathcal{L} \triangleq (L_1, L_2, L_3), \mathcal{I} \triangleq (I_1, I_2, I_3)$, and $L_1, L_2, L_3, I_1, I_2, I_3 \leftarrow \emptyset$.

*Phase 1 – Learning Leaf Clusters*

4: **for** each $x \in V^O \setminus A_{\text{obs}}$ **do**
5: $\quad$ **if** $\exists a \in A$ such that $d_{xa} + d_{aa'} = d_{xa'}$ for all $a' \in A \setminus \{a\}$ **then**
6: $\qquad$ **if** $\exists L \in \mathcal{L}$ such that $L_1 = a$ **then**
7: $\qquad\quad L_2 \leftarrow L_2 \cup \{x\}$
8: $\qquad$ **else**
9: $\qquad\quad L \triangleq (a, \{x\}, \emptyset)$
10: $\qquad\quad \mathcal{L} \leftarrow \mathcal{L} \cup L$
11: $\qquad$ **end if**
12: $\qquad V^O \leftarrow V^O \setminus \{x\}$
13: $\quad$ **end if**
14: **end for**
15: **Return** $\mathcal{L} = \{L : L_2 \in 2^{V^O \setminus A_{\text{obs}}}$ s.t. $L_2$ is separated from $A \setminus L_1$ by $L_1$ where $|L_1| = 1\}$.

*Phase 2 – Learning Internal Clusters*

16: **for** each $x \in V^O \setminus A_{\text{obs}}$ **do**
17: $\quad$ **for** each $\bar{A} \subset 2^A$ s.t. $|\bar{A}| > 1$ **do**
18: $\qquad$ **for** each pair $k, \ell \in \binom{\bar{A}}{2}$ **do**
19: $\qquad\quad$ **if** there exists a pair $(k, \ell)$ s.t. $d_{xk} + d_{k\ell} = d_{x\ell}$ or $d_{x\ell} + d_{\ell k} = d_{xk}$ **then**
20: $\qquad\quad$ **end if**
21: $\qquad\quad$ **Break**
22: $\qquad$ **end for**
23: $\quad$ **end for**
24: $\quad$ **if** $\exists$ a $I \in \mathcal{I}$ such that $I_1 = \bar{A}$ **then**
25: $\qquad I_2 \leftarrow I_2 \cup \{x\}$.
26: $\quad$ **else**
27: $\qquad I \triangleq (\bar{A}, \{x\}, \emptyset)$
28: $\qquad \mathcal{I} \leftarrow \mathcal{I} \cup I$
29: $\quad$ **end if**
30: **end for**
31: **Return** $\mathcal{I} = \{I : I_2 \in 2^{V^O \setminus A_{\text{obs}}}$ s.t. $I_2$ is separated from $A \setminus I_1$ by $I_1$ where $|I_1| > 1\}$.

---

**Procedure 5** NONCUTTEST

---

1: **Input:** A leaf cluster $L \in \mathcal{L}$ such that $|L_2| \geq 2$.
2: **Output:** A set $C_{\text{cut}}, C_{\text{non-cut}} \triangleq L_2 \setminus C_{\text{cut}}, L_3 \subseteq L_2$.
3: **Initialize:** $C_{\text{cut}}$ with $L_2$.
4: **for** each $x \in L_2$ **do**
5: $\quad$ **for** each pair $y, z \in L_2 \setminus \{x\}$ **do**
6: $\qquad$ Pick any arbitrary pair $\alpha_1, \alpha_2 \in V^O \setminus L_2$.
7: $\qquad U_i \triangleq \{x, y, \alpha_1\}$ and $U_j \triangleq \{x, z, \alpha_2\}$.
8: $\qquad$ **if** TIA$(U_i, U_j)$ is FALSE **then**

9: $\qquad$ **end if**
10: $\qquad$ **Break**
11: $\qquad C_{\text{cut}} \leftarrow C_{\text{cut}} \setminus \{x\}$ $\qquad\qquad$ ▷ $x$ is not a non-cut vertex.
12: $\quad$ **end for**
13: **end for**
14: **if** $(|C_{\text{cut}}|) > 1 \wedge \left( L_1 \notin A_{\text{obs}} \right)$ **then**
15: $\quad$ Pick an arbitrary vertex $a$ from $C_{\text{cut}}$ and set $L_3 \triangleq a$.
16: **end if**
17: **Return** $C_{\text{cut}}, C_{\text{non-cut}}$, and $L_3$.

---

---

**Subroutine 6** Partitioning and learning local edges (PaLE)

1: **Input:** The observed vertex set $V^O$, the collection of leaf clusters $\mathcal{L}$ and internal clusters $\mathcal{I}$.
2: **Output:** The vertex set $\mathcal{P}_{\text{op}}$, the articulation points $A_{\text{op}}$, and a subset $\mathcal{E}_{\text{leaf}}$ of the edge set $E_{\text{op}}$ for $\mathcal{T}_{\text{op}}$. Each element $E \in \mathcal{E}_{\text{leaf}}$ is an ordered quadruple such that $E_1, E_2 \subseteq V^O$, and $E_3 \in E_1, E_4 \in E_2$.
3: **Initialize** $\mathcal{P}_{\text{op}}, A_{\text{op}}, \mathcal{E}_{\text{leaf}} \leftarrow \emptyset$.
4: $A_{\text{cluster}} \triangleq \{c \in L_1 : c \in A_{\text{obs}}\}$
5: $\mathcal{P}_{\text{op}} \leftarrow \mathcal{P}_{\text{op}} \cup (A_{\text{obs}} \setminus A_{\text{cluster}}), A_{\text{op}} \triangleq A_{\text{obs}}.$

*Phase 1 – Partitioning and Local Edge Learning w.r.t. the Leaf Clusters*

6: **for** each $L \in \mathcal{L}$ s.t. $(|L_2| < 3) \wedge (L_1 \notin A_{\text{obs}})$ **do**  ▷ ancestor in the leaf cluster is not observed.
7:    Pick an arbitrary vertex $a \in L_2$, $L_3 \leftarrow a$, $L_2 \leftarrow L_2 \setminus \{a\}$, $A_{\text{op}} \leftarrow A_{\text{op}} \cup \{a\}$.
8:    $\mathcal{P}_{\text{op}} \leftarrow \mathcal{P}_{\text{op}} \cup L_2 \cup L_3$
9:    $\mathcal{E}_{\text{leaf}} \leftarrow \mathcal{E}_{\text{leaf}} \cup (L_2, L_3, L_2, L_3)$
10: **end for**
11: **for** each $L \in \mathcal{L}$ s.t. $(|L_2| \geq 3) \wedge (L_1 \notin A_{\text{obs}})$ **do** ▷ ancestor in the leaf cluster is not observed.
12:    Get $C_{\text{cut}}, C_{\text{non-cut}}$ and $L_3$ from NonCutTest($L$).
13:    $A_{\text{op}} \leftarrow A_{\text{op}} \cup L_3$
14:    Set $B \triangleq C_{\text{non-cut}} \cup L_3$ ▷ $C_{\text{non-cut}}$ and $L_3$ contains the non-cut vertices and cut vertex, respectively.
15:    $\mathcal{P}_{\text{op}} \leftarrow \mathcal{P}_{\text{op}} \cup B \cup \bigcup_{v \in C_{\text{cut}} \setminus L_3} \{v\}$ ▷ $C_{\text{cut}}$ can contain multiple cut vertices.
16:    $\mathcal{E}_{\text{leaf}} \leftarrow \mathcal{E}_{\text{leaf}} \cup \bigcup_{v \in C_{\text{cut}} \setminus L_3} (B, \{v\}, L_3, v)$
17: **end for**

18: **for** each $L \in \mathcal{L}$ s.t. $(|L_2| = 1) \wedge (L_1 \in A_{\text{obs}})$ **do** ▷ ancestor in the leaf cluster is observed.
19:    $\mathcal{P}_{\text{op}} \leftarrow \mathcal{P}_{\text{op}} \cup L_2 \cup L_3$
20:    $\mathcal{E}_{\text{leaf}} \leftarrow \mathcal{E}_{\text{leaf}} \cup (L_2, L_3, L_2, L_3)$
21: **end for**
22: **for** each $L \in \mathcal{L}$ s.t. $(|L_2| > 1) \wedge (L_1 \in A_{\text{obs}})$ **do** ▷ ancestor in the leaf cluster is observed.
23:    Get $C_{\text{cut}}, C_{\text{non-cut}}$ and $L_3$ from NonCutTest($L$).
24:    Set $B \triangleq C_{\text{non-cut}} \cup L_1$.
25:    $\mathcal{P}_{\text{op}} \leftarrow \mathcal{P}_{\text{op}} \cup B \cup \bigcup_{v \in C_{\text{cut}}} \{v\}$.
26:    $\mathcal{E}_{\text{leaf}} \leftarrow \mathcal{E}_{\text{leaf}} \cup \bigcup_{v \in C_{\text{cut}}} (B, \{v\}, L_3, v)$
27: **end for**

*Phase 2 – Partitioning w.r.t. the Internal Clusters*

28: **for** each $I \in \mathcal{I}$ **do**
29:    **for** each $i \in I_1$ **do**
30:       **if** $i \notin A_{\text{obs}}$ **then**
31:          Find the $L \in \mathcal{L}$ s.t. $L_1 = i$
32:          $I_3 \leftarrow I_3 \cup i, A_{\text{op}} \leftarrow A_{\text{op}} \cup \{i\}$
33:       **end if**
34:    **end for**
35:    $\mathcal{P}_{\text{op}} \leftarrow \mathcal{P}_{\text{op}} \setminus I_3$
36:    $B \triangleq \{I_2 \cup I_3\}$ and $\mathcal{P}_{\text{op}} \leftarrow \mathcal{P}_{\text{op}} \cup B$. ▷ $B$ is an internal non trivial block.
37: **end for**
38: **Return** $\mathcal{P}_{\text{op}}, A_{\text{op}}$, and a subset $\mathcal{E}_{\text{leaf}}$ of $E_{\text{op}}$.

---

set $\mathcal{P}_{\text{op}}$, (b) articulation points $A_{\text{op}}$, and a subset of the edge set $E_{\text{op}}$ for $\mathcal{T}_{\text{op}}$. The Subroutine 6 learns (a), (b), and (c) from both leaf clusters and internal clusters. In the following, we list all the possible cases of leaf clusters Subroutine 6 considered in learning $\mathcal{P}_{\text{op}}$ and $A_{\text{op}}$: Leaf clusters contains 1. At most two vertices with hidden ancestor, 2. More than two vertices with hidden ancestor, 3. One vertex with observed ancestor, and 4. More than one vertex with observed ancestor. For each $I \in \mathcal{I}$, Subroutine 6 checks whether $i \in A_{\text{obs}}$. If $i \notin A_{\text{obs}}$, then the subroutine finds the leaf cluster $L$ s.t. $L_1 \ni i$.

## A.2 Learning $E_{\text{op}}$ for $\mathcal{T}_{\text{op}}$

The next goal of NoMAD is to learn to learn the edge set $E_{\text{op}}$ for $A_{\text{op}}$. Precisely, NoMAD learns the neighbors of each articulation point in $A_{\text{op}}$. The learning of $E_{\text{op}}$ is divided into two steps: (a) Learn the neighbors of each articulation points (appears in Procedure 3), and (b) use the information obtained from (a) to construct $E_{\text{op}}$ (appears in Procedure 4).

## B Theory: Guaranteeing the Correctness of the NoMAD

In this section we will prove that NoMAD correctly learns the equivalence class. We star this section by restating Theorem 4.2 from Section 4.

**Theorem B.1.** *Consider a covariance matrix $\Sigma^*$ whose conditional independence structure is given by the graph $G$, and the model satisfies Assumption 4.1. Suppose that according to the problem setup in Section 3.1, we are given pairwise distance $d_{ij}$ of a vertex pair $(i, j)$ in the observed vertex set $V^O$, that is, $d_{ij} \triangleq -\log|\rho_{ij}|$ where $\rho_{ij} \triangleq \Sigma_{ij}^o / \sqrt{\Sigma_{ii}^o \Sigma_{jj}^o}$. Then, given the pairwise distance set $\{d_{ij}\}_{i,j \in V^o}$ as inputs, NoMAD outputs the equivalence class $[G]$.*

**Proof Outline.** We show that NoMAD correctly learns the equivalence class by showing that it can correctly learn $\mathcal{T}_{\text{op}}$. Given this, and using Definition 3.2, the entire equivalence class can be readily generated. We show that NoMAD learns $\mathcal{T}_{\text{op}}$ correctly by proving that (a) the vertex set $\mathcal{P}_{\text{op}}$, (b) the articulation points $A_{\text{op}}$, and (c) the edge set $E_{\text{op}}$ are learnt correctly. Following is the outline for (a) and (b). From Section 3.3, it is clear that NoMAD succeeds in finding the ancestors, which is the first step, provided the TIA tests succeed (established in Lemma B.7).Then, Proposition B.13 establishes that NoMAD learns $\mathcal{P}_{\text{op}}$ and $A_{\text{op}}$ correctly. The proof correctness of this step crucially depends on identifying the non-cut vertices (c.f. Lemma B.12).

Then, for establishing the correctness of NoMAD in learning $E_{\mathrm{op}}$, NoMAD learns the neighbor articulation points of each articulation point. Proposition B.14 shows that NoMAD correctly learns $E_{\mathrm{op}}$.

**Lemma B.1.** *Let $G$ be a graph on vertex set $V$, and $\mathcal{T}_{\mathrm{AST}}(G)$ be the corresponding articulated set tree of $G$. Then, $\mathcal{T}_{\mathrm{AST}}(G)$ is a tree.*

*Proof.* We prove $\mathcal{T}_{\mathrm{AST}}(G)$ is a tree by showing that $\mathcal{T}_{\mathrm{op}}$ is connected and acyclic. Suppose on the contrary that $\mathcal{T}_{\mathrm{AST}}(G)$ contains a cycle $B'$. Then, $B'$ is a non-trivial block in $G$ with no cut vertex. This would contradict the maximality of the non-trivial blocks contained in the cycle $B'$. Hence, any cycle is contained in a unique non-trivial block in $\mathcal{T}_{\mathrm{AST}}(G)$. We now show that $\mathcal{T}_{\mathrm{op}}$ is connected. Recall that vertices in $\mathcal{T}_{\mathrm{AST}}(G)$ can either be a non-trivial block or a singleton vertices not part of any non-trivial block. Consider any vertex pair $(u, v)$ in $\mathcal{T}_{\mathrm{AST}}(G)$. We will find a path from $u$ to $v$. Suppose that $u$ and $v$ are non-singletons, and associated with non-trivial blocks $B_u$ and $B_v$ respectively. Since, $G$ is connected, $\exists$ a path between the articulation points of $B_u$ and $B_v$. Hence, $u$ and $v$ are connected in $\mathcal{T}_{\mathrm{AST}}(G)$. The other cases where one of them is a singleton vertex or both are singleton vertices follows similarly.

$\square$

We now show that NoMAD correctly learns $[G]$. For the graph $G$ on a vertex set $V$, let $G^{\mathrm{J}} = (V^{\mathrm{J}}, E^{\mathrm{J}})$ be defined as in Subsection 3.1. Let $A^{\mathrm{J}}$ be the set of ancestors in $G^{\mathrm{J}}$. Recall that NoMAD only observes samples from a subset $V^{\mathrm{O}} \subseteq V^{\mathrm{J}}$ of vertices. NoMAD uses $\{d_{ij}\}_{i,j \in V^{\mathrm{O}}}$ to learn $\mathcal{T}_{\mathrm{op}}$, which in turn will output $[G]$. Hence, each theoretical section first states a result of $G^{\mathrm{J}}$ *assuming* that the pair $(V^{\mathrm{J}}, E^{\mathrm{J}})$ is known.

**Correctness in Learning $\mathcal{P}_{\mathrm{op}}$ and $A_{\mathrm{op}}$.** We first establish that NoMAD correctly identifies ancestors in $G^{\mathrm{J}}$. In the following, we first identify the vertices in $G$ which are ancestors in $G^{\mathrm{J}}$. Then, in Lemma B.5, we show the existence of at least two vertex triplets for each ancestor in $G^{\mathrm{J}}$. Finally, in Proposition B.10, we show that Subroutine 3 correctly identifies the star triplets in $G^{\mathrm{J}}$. We start with introducing $uw$-separator.

**Definition B.2** ($uw$-separator). *Consider an arbitrary pair $u, w \in V$ in the graph $G$. We say $v \in V \setminus \{u, w\}$ is a $uw$- separator in $G$ if and only if any path $\pi \in \mathcal{P}_{uw}$ contains $v$.*

**Lemma B.3.** *A vertex $a \in V^{\mathrm{J}}$ is an ancestor in $G^{\mathrm{J}}$ if and only if $a$ is an $uw$- separator in $G$, for some $u, w \in V$.*

*Proof.* ($\Rightarrow$) Suppose $a \in V^{\mathrm{J}}$ is an ancestor in $G^{\mathrm{J}}$. Then, we show that $a$ is an $uw$- separator in $G$. Fix a triplet $T \triangleq \{a_1^e, a_2^e, a_3^e\} \in \mathcal{V}_a$, where $\mathcal{V}_a \triangleq$ collection of all triplets with ancestor $a$ in $G^{\mathrm{J}}$. Then, any path $\pi \in \mathcal{P}_{a_i^e a_j^e}$ contains $a$ in $G^{\mathrm{J}}$, for $i, j = 1, 2$, and 3. Thus, $a_i^e \perp\!\!\!\perp a_j^e | a$, and $a$ is an $uw$-separator with $u = a_i^e$, and $w = a_j^e$. Furthermore, for a joint graph following is true for any vertex $u$ and its corresponding noisy samples $u^e$: $u^e \perp\!\!\!\perp v | u$ for all $v \in V^{\mathrm{J}} \setminus \{u, u^e\}$. Hence, we can conclude that $a_i \perp\!\!\!\perp a_j \mid a$, and $a$ is an $uw$-separator in $G$.

($\Leftarrow$) Suppose that $\exists u, w \in V$ for which $a \in V$ is an $uw$- separator in $G$. Then, we show that $v$ is an ancestor in $G^{\mathrm{J}}$ by constructing a triplet $T$ with ancestor $a$. This construction directly follows from Definition B.2 and Definition 3.4.

$\square$

**Lemma B.4.** *Let $V_{\mathrm{cut}}$ be the set of all cut vertices in $G$. Then, there does not exist any pair $u, w \in V$ such that $b \in V \setminus V_{\mathrm{cut}}$ is an $uw$- separator in $G$.*

*Proof.* Let $b \in V \setminus V_{cut}$. Then, notice that $b$ can be either (1) a non-cut vertex of a non-trivial block or a (2) leaf vertex in $G$. For (1), by the definition of a block, any non-cut vertex ceases to be a $uw$-separator for any $u \neq b$ and $w \neq b$ in $V$. For (2), since $b$ is a leaf vertex, its degree is one, and hence, cannot be a $uw$-separator for any $u \neq b$ and $w \neq b$ in $V$. $\square$

**Lemma B.5.** *Let $A^{\mathrm{J}}$ be the set of all ancestors in $G^{\mathrm{J}}$. Then, for each $a \in A^{\mathrm{J}}$, there exists at least two triplets $U, W \in \binom{V^{\mathrm{O}}}{3}$ for which $a$ is the ancestor in $G^{\mathrm{J}}$.*

*Proof.* We construct two triplets for any ancestor in $G^J$. Lemma B.4 states that only a cut-vertex in $G$ is an ancestor in $G^J$. First, let $c$ be a cut-vertex of a non-trivial block $B$ in $G$. Pick any two non-cut vertices $x, y \in B \setminus \{c\}$. Then, consider the following two triplets in $V^O$ : $\{x^e, c^e, \alpha_1^e\}$ and $\{y^e, c^e, \alpha_2^e\}$, where $\alpha_1, \alpha_2 \in V \setminus B$. Then both $\{x^e, c^e, \alpha_1^e\}$ and $\{y^e, c^e, \alpha_2^e\}$ share the ancestor $c$ in $G^J$. Now, let $c$ be a cut vertex which is not in any non-trivial block. Consider two blocks $B_i$ and $B_j$ such that $B_i \perp\!\!\!\perp B_j \mid c$. Then, consider the following pair: $\{i_1, c, j_1\}$ and $\{i_2, c, j_2\}$ s.t. $i_1, i_2 \in B_i$ and $j_1, j_2 \in B_j$. Notice that $\{i_1^e, c^e, j_1^e\}$ and $\{i_2^e, c^e, j_2^e\}$ in $\binom{V^O}{3}$ share the ancestor $c$ in $G^J$. Finally, if $G$ is a tree on three vertices, then $G$ has an unique ancestor.

$\square$

**Claim 1.** *Let (a) $\{i, j, k\}$ be a vertex triple in $G$, and (b) $i^e$ be the corresponding noisy counterpart of $i$. Then, $j$ separates $i$ and $k$ if and only if $j$ separates $i^e$ and $k$ in combined graph $G^J$*

*Proof.* The forward implication directly follows from the construction of joint graph. For the reverse implication suppose that in $G^J$, $i^e \perp\!\!\!\perp k \mid j$. We show that this implies $i \perp\!\!\!\perp k \mid j$ and $k$ in $G$. Suppose on the contrary that $i \not\perp\!\!\!\perp k \mid j$. That means $\exists$ a path $\pi$ between $i$ and $k$ that does not contain $j$. Now, notice that $\pi \cup \{i, i^e\}$ is a valid path between $i^e$ and $k$ in $G^J$ that does not contain $j$, and it violates the hypothesis.

$\square$

The following lemma relates an observed ancestor in a triplet $T$ with the remaining pair.

**Lemma B.6.** *Suppose that a triplet $T \in \binom{V^O}{3}$ is a star triplet in $G^J$. A vertex $v \in T$ is an $uw-$ separator for $u, w \in T \setminus v$ if and only if $v$ is the ancestor of $T$.*

*Proof.* Suppose that a vertex $v \in T$ is an $uw-$ separator for $u, w \in T \setminus v$. We show that $v$ is an ancestor. As $v$ is an $uw-$ separator, i.e., $u \perp\!\!\!\perp w \mid v$. Suppose on the contrary that $v' \neq v$ is the ancestor of $T$ in $G^J$. We show that $v$ is not an $uw-$ separator for $u, w \in T \setminus v$. As $v'$ is the ancestor of $T$, $u \perp\!\!\!\perp w \mid v'$. (according to Definition 3.4). This contradicts the hypothesis that $u \perp\!\!\!\perp w \mid v$. Thus, $v$ and $v'$ are identical. Therefore, $v$ is the ancestor of $\{u, v, w\}$. The reverse implication follows from Definition 3.4. $\square$

We will now prove the correctness of the TIA test. We proceed with the following claim.

**Claim 2.** *Suppose that $U$ and $W \in \binom{V^O}{3}$ are star triplets with non-identical ancestors $r_u$ and $r_w$, resp. Then, there exists a vertex $u \in U$ and a pair, say $w_2, w_3 \in W$, such that all paths $\pi \in \mathcal{P}_{uw_i}$ for $i = 1, 2$ contain both $r_u$ and $r_w$.*

*Proof.* Without loss of generality, let $W = \{w_1, w_2, w_3\}$. We prove this claim in two stages. In the first stage, we show that for each vertex $u \in U$ there exists at least a pair $w_2, w_3 \in W$ such that $u \perp\!\!\!\perp \{w_2, w_3\} \mid r_w$. Then, in the next stage we show that there exists a vertex $u \in U$ such that $u \perp\!\!\!\perp r_w \mid r_u$. For the first part, suppose on the contrary that there exists a vertex $u \in U$ and a pair $w_2, w_3 \in W$ such that there exists a path $\pi_2 \in \mathcal{P}_{uw_2}$ and a path $\pi_3 \in \mathcal{P}_{uw_3}$ such that $r_w \notin \pi_2$ and $r_w \notin \pi_3$. Then, one can construct a path between $w_2$ and $w_3$ that does not contain $r_w$, which violates the hypothesis that $W$ is a star triplet. Now, in the next step of proving the claim, we show that there exists a vertex $u \in U$ such that $u \perp\!\!\!\perp r_w \mid r_u$. Now, suppose that for all $u \in U$ there exists a path between $u$ and $r_w$, that does not contain $r_u$. We will next show that this implies there has to be a path between $u_1$ and $u_2$ ($u_1, u_2 \in U$) that does not include $r_u$. We will show this constructively. Let $s$ be the last vertex in the path $\pi_{u_1 r_w}$ that is also contained in $\pi_{u_2 r_w}$. Note that $\pi_{u_1 s}$ and $\pi_{u_2 s}$ are valid paths in the graph, and that their concatenation is a valid path between $u_1$ and $u_2$. This proves that $u_1$ and $u_2$ are connected by a path that is not separated by $r_u$, and hence contradicting the hypothesis that $U$ is a star triplet. Finally, let $u' \in U$ be the vertex for which $u' \perp\!\!\!\perp r_w \mid r_u$. Then, there exists a triplet $\{u', w_2, w_3\}$ such that both $r_u$ and $r_w$ separates $u'$ and $w_2$, and both $r_u$ and $r_w$ separates $u'$ and $w_3$. $\square$

Using Claim 2, we now show the correctness of our TIA test. Recall that the TIA $(U, W)$ accepts triplets $U, W \in \binom{V^O}{3}$, and returns TRUE if and only if $U$ and $W$ share an ancestor in $G^J$. Also recall the following assumption: Let $U, W \in \binom{V}{3} \setminus \mathcal{V}_{\text{star}} \cup \mathcal{V}_{\text{sep}}$. Then, (i) there are no vertices $x \in U$ and $a \in W$ that satisfy

$d_x^U + d_a^W = d_{xa}$, and (ii) there does not exist any vertex $r \in V$ and $x \in U$ for which the distance $d_{xr}$ satisfies relation in equation 1.

**Lemma B.7.** *(**Correctness of TIA test**)* *Fix any two vertex triplets $U \neq W \in \binom{V^o}{3}$. TIA$(U,W)$ returns* TRUE *if and only if $U$ and $W$ are star triplets in $G^J$ with an identical ancestor $r \in V$.*

*Proof.* From Subroutine 2 returning TRUE is same as checking that for all $x \in U$, there exist at least two vertices $y, z \in W$ such that both of the following hold

$$d_x^U + d_y^W = d_{xy}, \tag{2}$$
$$d_x^U + d_z^W = d_{xz}. \tag{3}$$

Suppose that $U$ and $W$ are star triplets with an identical ancestor $r \in V$. We prove by contradiction. Let $a \in U$ and assume that there is *at most* one vertex $x \in V$ such that $d_a^U + d_x^W = d_{ax}$. Therefore, one can find two vertices $y_1, y_2 \in V$ such that

$$d_a^U + d_{y_i}^W \neq d_{ay_i}, \quad i = 1, 2. \tag{4}$$

However, from our hypothesis that $U$ and $W$ are star triplets with the common ancestor $r$, we know that $d_a^U = d_{ar}$ and $d_{y_i}^W = d_{ry_i}$, for $i = 1, 2$. This, along with equation 4, implies that $r$ does not separate $a$ from $y_1$ or $y_2$. For $i = 1, 2$, let $\pi_{ay_i}$ be the path between $a$ and $y_i$ that does not include $r$. We will next show that this implies there has to be a path between $y_1$ and $y_2$ that does not include $r$. We will show this constructively. Let $s$ be the last vertex in the path $\pi_{ay_1}$ that is also contained in $\pi_{ay_2}$. Note that $\pi_{y_1s}$ and $\pi_{sy_2}$ are valid paths in the graph, and that their concatenation is a valid path between $y_1$ and $y_2$. This proves that $y_1$ and $y_2$ are connected by a path that is not separated by $r$, and hence contradicting the first hypothesis.

For the reverse implication, we do a proof by contrapositive. Fix two triplets $U$ and $W$. Suppose that $U$ and $W$ are *not star triplets with an identical ancestor* in $G^J$. We will show that this implies that there exists at least one vertex in $U$ for which no pair in $W$ satisfies both Eq. equation 2 and Eq. equation 3. To this end, we will consider all three possible configurations for a triplet pair $U$ and $W$ where they are not star triplets with an identical ancestor in $G^J$: 1. $U$ and $W$ are star triplets with a non-identical ancestor in $G^J$, 2. Both $U$ and $W$ are non-star triplets in $G^J$, 3. $U$ is a star triplet and $W$ is a non-star triplet in $G^J$. Then, for each configuration, we will show that there exists at least a vertex $x \in U$ for which no pair in $W$ satisfies both Eq. equation 2 and Eq. equation 3.

$U$ **and** $W$ **are star triplets with non-identical ancestors.** Let $U$ and $W$ be two star triplets with two ancestor $r_u$ and $r_w$, respectively, such that $r_u \neq r_w$. As $U$ and $W$ are star triplets, $d_x^U$ and $d_y^W$ returns the distance from their corresponding ancestors $d_{xr_u}$ for all $x \in U$, and $d_{yr_w}$ for all $y \in W$, respectively. Now, according to the Claim 2, there exists a vertex triplet, say $\{u, w_1, w_2\}$ w.l.o.g., where $u \in U$ and $w_1, w_2 \in W$ such that $u$ is separated from $w_i$ for $i = 1, 2$ by both $r_u$ and $r_w$. Furthermore, the same $u$ identified above is separated from $r_w$ by $r_u$. This implies that $d_{uw_i} = d_{ur_u} + d_{r_u w_i} = d_{ur_u} + d_{r_u r_w} + d_{r_w w_i}$ for $i = 1, 2$. As we know that $r_u$ and $r_w$ are not identical, $d_{r_u r_w} \neq 0$, which implies that $d_{uw_i} \neq d_{ur_u} + d_{r_w w_i}$, where $i = 1, 2$. Thus we conclude the proof for the first configuration by showing that there exists a vertex $u \in U$ and a pair $w_1, w_2 \in W$ such that the identities in equation 2 and equation 3 do not hold.

$U$ **is a star triplet and** $W$ **is a non-star triplet in** $G^J$**.** We show that there exists a triplet $\{y, a, b\}$ where $y \in U$ and $a, b \in W$ such that identities in equation 2 and equation 3 do not hold. Let $W$ be a non-star triplet, and $U$ be a star triplet with the ancestor $r \in V$ in $G^J$. Now, as $U$ is a star triplet, $d_x^U$ returns the distance from its ancestor $d_{xr}$ for all $x \in U$. Suppose that there exists a vertex pair $x \in U$ and $a \in W$ for which $d_{xr} + d_a^W = d_{ax}$. We know that for a non-star triplet $W$, the computed distance $d_a^W \neq d_{ar}$ for any $a \in W$ from Assumption 4.1. Thus, for the pair $\{x, a\}$, $d_{xr} + d_{ar} \neq d_{xa}$. This implies from the Fact 1 that $x \not\perp\!\!\!\perp a \mid r$. Similarly, we can conclude that $x \not\perp\!\!\!\perp b \mid r$. Then, $y \perp\!\!\!\perp a \mid r$ and $y \perp\!\!\!\perp b \mid r$. Otherwise, one can construct a path between $y$ and $x$ that does not contain $r$ which violates the assumption that $U \ni x, y$ is a star triplet with ancestor $r$. As $y \perp\!\!\!\perp a \mid r$ and $y \perp\!\!\!\perp b \mid r$, using the Fact 1 we have that $d_{yr} + d_{ra} = d_{ya}$ and $d_{yr} + d_{rb} = d_{yb}$. As $a \in W$, and $d_{ar} \neq d_a^W$, thus, $d_{yr} + d_a^W \neq d_{ya}$. Similarly, for the pair $\{y, b\}$, we have that $d_{yr} + d_b^W \neq d_{yb}$. Thus, for the triplet $\{y, a, b\}$, the identities in Eq. equation 2 and equation 3 do not hold.

$U$ **and** $W$ **are both non-star triplets in** $G^{\mathbf{j}}$**.** The proof for this configuration follows from the Assumption 4.1.

Notice that these three cases combined proves that the TIA test returns TRUE if and only if the triplets considered are both start triplets that share a common ancestor. □

Now recall that the first phase of Subroutine 3 identifies the star triplets in $G^{\text{J}}$, the observed ancestors in $G^{\text{J}}$, and outputs a set $A_{\text{hid}}$ such that $|A_{\text{hid}}|$ equals to the number of hidden ancestors in $G^{\text{J}}$. Formally, the result is as follows.

**Proposition B.8** (Correctness of Subroutine 3 in identifying ancestors)**.** *Given the pairwise distances* $\{d_{ij}\}_{i,j \in V^o}$*, Subroutine 3 correctly identifies (a) the star triplets in* $G^{\text{J}}$*, (b) the observed ancestors in* $G^{\text{J}}$*, and (b) introduces a set* $A_{hid}$ *such that* $|A_{hid}|$ *equals to the number of hidden ancestors in* $G^{\text{J}}$

*Proof.* Combining Lemma B.5 and Lemma B.7, we prove that the Subroutine 3 successfully cluster the star triplets in $G^{\text{J}}$. Then, it partitions $\mathfrak{V}$ into $\mathfrak{V}_{\text{obs}}$ and $\mathfrak{V}_{\text{hid}}$ s.t. following is true: for any triplet collection $\mathcal{V}_i \in \mathfrak{V}_{\text{obs}}$ ($\mathcal{V} \in \mathfrak{V}_{\text{hid}}$ resp.), the ancestor of the triplets in $\mathcal{V}_i$ is observed (hidden resp.) Finally, Subroutine 3 outputs a set $A_{\text{hid}}$ s.t. $|A_{\text{hid}}| = |\mathfrak{V}_{\text{hid}}|$. □

We show the correctness of Subroutine 3 in learning (a) $\{d_{ij}\}_{i^e \in V^o, j \in A_{\text{hid}}}$, and (b) $\{d_{ij}\}_{i,j \in A_{\text{hid}}}$.

**Claim 3.** *Fix any* $a \in A^{\text{J}}$*, where* $A^{\text{J}} =$ *set of all ancestors. Let* $\mathcal{V}_a$ *be the collection of vertex triplets which shares common ancestor. Then, any* $i \in A^{\text{J}}$ *is s.t. at least one triplet in* $\mathcal{V}_a$*.*

*Proof.* Fix any ancestor $a \in A^{\text{J}}$. Construct a triplet $T_i$ for a fixed vertex $i \neq a \in V^{\text{J}}$ s.t. $a$ is the ancestor of $T_i$ in $G^{\text{J}}$. From Lemma B.4: $a$ is a cut vertex in $G$. Thus, fixing $i$ and $a$, find another vertex $w \in V$ such that $a$ separates $i$ and $w$ in $G$. Hence, from Lemma B.3 we can conclude the following: $a$ is the ancestor for the triplet $T_i \triangleq \{i, a, w\}$ in $G^{\text{J}}$. □

**Claim 4.** *Let* $U_i$ *and* $U_j$ *be both star triplets with ancestor* $i$ *and* $j$ *respectively, and* $i \neq j$*. Let* $x \in U_i$ *and* $y \in U_j$ *be a vertex pair such that* $x \perp\!\!\!\perp y | i$ *and* $x \not\perp\!\!\!\perp y | j$ *Then,* $x \not\perp\!\!\!\perp i | j$*.*

*Proof.* $x \perp\!\!\!\perp y | i$ implies any path between $x$ and $y$ contains $i$. $x \not\perp\!\!\!\perp y | j$ implies $\exists$ a path $\pi$ between $x$ and $y$ that does not contain $j$. Notice that, the path $\pi$ contains $i$. As $\pi$ contains both $x$ and $i$, $\exists$ a path between $x$ and $i$ which does not contain $j$. Hence, $x \not\perp\!\!\!\perp i | j$. □

**Lemma B.9.** *For any pair of distinct ancestors* $i, j \in A^{\text{J}}$*, pick arbitrary triplets* $U_i \in \mathcal{V}_i$ *and* $U_j \in \mathcal{V}_j$*. Define the set* $D(U_i, U_j)$ *as follows:*

$$\Delta(U_i, U_j) \triangleq \left\{ d_{xy} - (d_x^{U_i} + d_y^{U_j}) : x \in U_i, y \in U_j \right\} \tag{5}$$

*The most frequent element in* $\Delta(U_i, U_j)$*, that is,* $\text{mode}(\Delta_{ij})$ *is the true distance* $d_{ij}$ *with respect to* $G^{\text{J}}$*.*

*Proof.* To aid exposition, we suppose that $U_i = \{x_1, x_2, x_3\}$ and $U_j = \{y_1, y_2, y_3\}$. We also define for any $x \in U_i$ and $y \in U_j$: $\Delta(x, y) \triangleq d_{xy} - \left( d_x^{U_i} + d_y^{U_j} \right)$. Observe that according to the Claim 2, for two start triplets $U_i, U_j \in \binom{V^o}{3}$ with non-identical ancestors, there exist a vertex, say $x_1 \in U_i$ and a pair, say $y_1, y_2 \in U_j$ such that following is true: $x_1 \perp\!\!\!\perp y_i | i$ and $x_1 \perp\!\!\!\perp y_i | j$ for $i = 1, 2$ Furthermore, the same $x_1$ (identified above) is separated from $j$ by $i$, that is, $x_1 \perp\!\!\!\perp j | i$. This similar characterization is also true for a vertex triplet where one vertex is from $U_j$ and a pair from $U_i$. Now observe that

$$\Delta(x_1, y_1) = d_{x_1 y_1} - d_{x_1 i} - d_{y_1 j} = d_{x_1 j} + d_{y_1 j} - d_{x_1 i} - d_{y_1 j} = d_{x_1 j} - d_{x_1 i} = d_{ij}.$$

Similarly, it can be checked that $\Delta(x_1, y_2) = d_{ij}$. The similar calculation can be shown for the other triplet (where one vertex is from $U_j$ and a pair from $U_i$). In other words, we have demonstrated that 4 out of the 9 total distances in $D(U_i, U_j)$ are equal to $d_{ij}$. All that is left to be done is to show that no other value can have a multiplicity of four or greater.

Now, our main focus is to analyze the five remaining distances, i.e., $\Delta(x_3, y_3)$, $\Delta(x_3, y_1)$, $\Delta(x_3, y_2)$, $\Delta(x_1, y_3)$, and $\Delta(x_2, y_3)$, for two remaining configurations: (a) $x_3$ is separated from $y_3$ by only one vertex in $\{i, j\}$, and (b) $x_3 \not\perp\!\!\!\perp y_3|i$ *and* $x_3 \not\perp\!\!\!\perp y_3|j$. For configuration (a), consider without loss of generality that $x_3 \perp\!\!\!\perp y_3|i$ and $x_3 \not\perp\!\!\!\perp y_3|j$. Then, according to Claim 4, we have the following two possibilities:

1. $x_3 \perp\!\!\!\perp y_3|i, x_3 \not\perp\!\!\!\perp y_3|j$, *and* $x_3 \perp\!\!\!\perp j|i$: As $x_3 \not\perp\!\!\!\perp y_3|j$, it must be the case that $x_3 \perp\!\!\!\perp y_\nu|j$ for $\nu = 1, 2$. Otherwise, one can construct a path between $y_1$ and $y_\nu$ which does not contain $j$, and that violates the hypothesis that $U_j = \{y_1, y_2, y_3\}$ is a star triplet. Next, notice that in this set up, $x_3 \perp\!\!\!\perp j|i$. Now, notice the following:

$$\Delta(x_3, y_\nu) = d_{x_3 y_\nu} - d_{x_3 i} - d_{y_\nu j} = d_{x_3 j} + d_{y_\nu j} - d_{x_3 i} - d_{y_\nu j} = d_{x_3 i} + d_{ij} - d_{x_3 i} = d_{ij}.$$

Therefore, for this set up, six distances are equal to $d_{ij}$.

2. $x_3 \perp\!\!\!\perp y_3|i, x_3 \not\perp\!\!\!\perp y_3|j$, *and* $x_3 \not\perp\!\!\!\perp j|i$: As $x_3 \not\perp\!\!\!\perp y_3|j$, it must be the case that $x_3 \perp\!\!\!\perp y_\nu|j$ for $\nu = 1, 2$. Otherwise, one can construct a path between $y_1$ and $y_\nu$ which does not contain $j$, and that violates the hypothesis that $U_j = \{y_1, y_2, y_3\}$ is a star triplet. $\Delta(x_3, y_\nu) = d_{x_3 y_\nu} - d_{x_3 i} - d_{y_\nu j} = d_{x_3 j} + d_{y_\nu j} - d_{x_3 i} - d_{y_\nu j} = d_{x_3 j} - d_{x_3 i}$. Now, $d_{x_3 j} - d_{x_3 i}$ equals to $d_{ij}$ implies that $x_3 \perp\!\!\!\perp j|i$ which contradicts the setup. Therefore, $\Delta(x_3, y_\nu)$ not equals to $d_{ij}$. Therefore, for this set up, even if three remaining distances are equal, correct $d_{ij}$ will be chosen.

In the following we will analyze the distance between $\Delta(x_3, y_3)$ and $\Delta(x_3, y_\nu)$ using the following assumption common in graphical models literature: For any vertex triplet $i, j, k \in \binom{V^o}{3}$, if $i \not\perp\!\!\!\perp j|k$, then $|d_{ij} - d_{ik} - d_{jk}| > \gamma$.

$$\Delta(x_3, y_3) - \Delta(x_3, y_\nu) = d_{x_3 y_3} - d_{x_3 i} - d_{y_3 j} - d_{x_3 y_\nu} + d_{x_3 i} + d_{y_\nu j},$$
$$= d_{x_3 i} + d_{y_3 i} - d_{x_3 i} - d_{y_3 j} - d_{x_3 j} - d_{y_\nu j} + d_{x_3 i} + d_{y_\nu j} = d_{y_3 i} + d_{x_3 i} - d_{y_3 j} - d_{x_3 j} = d_{x_3 y_3} - d_{y_3 j} - d_{x_3 j}.$$

Now, as $x_3 \not\perp\!\!\!\perp y_3|j$ according to Assumption 5.2, $|\Delta(x_3, y_3) - \Delta(x_3, y_\nu)| > \gamma$ for $\nu = 1, 2$. For configuration (b), we analyze the five remaining distances, i.e., $\Delta(x_3, y_3)$, $\Delta(x_3, y_1)$, $\Delta(x_3, y_2)$, $\Delta(x_1, y_3)$, and $\Delta(x_2, y_3)$, and show that these five distances can not be identical which in turn will prove the lemma.

$x_3 \not\perp\!\!\!\perp y_3|i$ *and* $x_3 \not\perp\!\!\!\perp y_3|j$. For this configuration we note the following two observations:

O1 As $x_3 \not\perp\!\!\!\perp y_3|j$, it must be the case that $x_3 \perp\!\!\!\perp y_\nu|j$ for $\nu = 1, 2$. Otherwise, one can construct a path between $y_1$ and $y_\nu$ which does not contain $j$, and that violates the hypothesis that $U_j = \{y_1, y_2, y_3\}$ is a star triplet.

O2 Similarly, as $x_3 \not\perp\!\!\!\perp y_3|i$, it must be the case that $x_\nu \perp\!\!\!\perp y_3|i$ for $\nu = 1, 2$. Otherwise, one can construct a path between $x_1$ and $x_\nu$ which does not contain $i$, and that violates the hypothesis that $U_i = \{x_1, x_2, x_3\}$ is a star triplet.

Recall that our goal for configuration (b) is to analyze the distances $\Delta(x_3, y_3)$, $\Delta(x_3, y_1)$, $\Delta(x_3, y_2)$, $\Delta(x_1, y_3)$, and $\Delta(x_2, y_3)$. We start with the distance pair $\Delta(x_3, y_\nu)$ and $\Delta(x_3, y_3)$ for $\nu = 1, 2$.

$$\Delta(x_3, y_\nu) \overset{(a)}{=} d_{x_3 j} + d_{y_\nu j} - d_{x_3 i} - d_{y_\nu j} = d_{x_3 j} - d_{x_3 i},$$

where (a) follows from the O1. Furthermore, the distance $\Delta(x_3, y_3) = d_{x_3 y_3} - d_{y_3 j} - d_{x_3 i}$. Now, $\Delta(x_3, y_\nu)$ equals to $\Delta(x_3, y_3)$ implies that $d_{x_3 j} - d_{x_3 i} = d_{x_3 y_3} - d_{y_3 j} - d_{x_3 i}$ which is equivalent to saying that $d_{x_3 j} + d_{y_3 j}$ equals to $d_{x_3 y_3}$. Then, $d_{x_3 j} + d_{y_3 j} = d_{x_3 y_3}$ will imply $x_3 \perp\!\!\!\perp y_3|j$ – which contradicts the hypothesis of the configuration that $x_3 \not\perp\!\!\!\perp y_3|j$.

Thus, $\Delta(x_3, y_3)$ is not equal to $\Delta(x_3, y_\nu)$ for $\nu = 1, 2$. (based on O1). Similarly, (based on the O2) $\Delta(x_3, y_3)$ is not equal to $\Delta(x_\nu, y_3)$ for $\nu = 1, 2$. Thus, the distance $\Delta(x_3, y_3)$ is not equal to any of the following distances: $\Delta(x_3, y_1), \Delta(x_3, y_2), \Delta(x_1, y_3)$, and $\Delta(x_2, y_3)$.

Now all that remains to prove the lemma is to show that the 4 (remaining) distances $\Delta(x_3, y_1)$, $\Delta(x_3, y_2)$, $\Delta(x_1, y_3)$, and $\Delta(x_2, y_3)$ are not identical. To this end, we analyze two distances: $\Delta(x_1, y_3)$ and $\Delta(x_3, y_1)$. First notice from the O2 that $\Delta(x_1, y_3) = d_{x_1 i} + d_{x_3 i} - d_{x_3 i} - d_{y_3 j}$ equals to $d_{y_3 i} - d_{y_3 j}$, and $\Delta(x_3, y_1) = d_{x_3 j} + d_{y_3 j} - d_{x_3 i} - d_{y_3 j}$ equals to $d_{x_3 j} - d_{x_3 i}$. As neither $i$ nor $j$ is separating $x_3$ from $y_3$, the event that $d_{y_3 i} - d_{y_3 j}$ equals to $d_{x_3 j} - d_{x_3 i}$ happens only on a set of measure zero. We end this proof by computing the distance between $\Delta(x_3, y_3)$ and $\Delta(x_3, y_\nu)$.

$$\Delta(x_3, y_3) - \Delta(x_3, y_\nu) = d_{x_3 y_3} - d_{y_3 j} - d_{x_3 i} - d_{x_3 j} - d_{y_\nu j} + d_{x_3 i} + d_{y_\nu j} = d_{x_3 y_3} - d_{y_3 j} - d_{x_3 j}.$$

Now, as $x_3 \not\perp\!\!\!\perp y_3 | j$, according to Assumption 5.2, $|\Delta(x_3, y_3) - \Delta(x_3, y_\nu)| > \gamma$ for $\nu = 1, 2$. □

**Lemma B.10** (Correctness in *extending the distances.*). *Given $\{d_{ij}\}_{i,j \in V^\circ}$, Subroutine 3 correctly learns (a) $\{d_{ij}\}_{i,j \in V^\circ \cup A}$ and (b) $\{d_{ij}\}_{i,j \in A_{hid}}$, where $A \triangleq A_{obs} \cup A_{hid}$.*

*Proof.* Follows directly from Claim 3 and Lemma B.9. □

**Lemma B.11** (Correctness in learning clusters). *Subroutine 4 correctly learns leaf clusters and internal clusters.*

*Proof.* As the distances $\{d_{ij}\}_{i,j \in A_{hid}}$ and $\{d_{ij}\}_{i,j \in V^\circ \cup A_{hid}}$ are learned correctly by Subroutine 3, where $V^\circ$ and $A_{hid}$ is the set of observed vertices, and hidden ancestors, respectively, the correctness of learning the leaf clusters and internal clusters follows from Fact 1. □

**Lemma B.12.** *Let $L \subset 2^V$ be a subset of vertices in $G$ s.t. only noisy samples are observed from the vertices in $L$. Then, $\exists$ a vertex $v \in L$, where $v \in V_{cut}$, s.t. $v$ separates $L \setminus \{v\}$ from the remaining vertices $v' \in V_{cut} \setminus \{v\}$ Let $L^e$ be the noisy counterpart of $L$. The noiseless counterpart of $x^e$ is a non-cut vertex if and only if $\exists$ at least a pair $y^e, z^e \in L^e \setminus \{x^e\}$ such that $TIA(\{x^e, y^e, \alpha_1^e\}, \{x^e, z^e, \alpha_2^e\})$ returns FALSE, where $\alpha_1^e, \alpha_2^e \in V^J \setminus L^e$.*

*Proof.* ($\Rightarrow$) Suppose that $x$ is a non-cut vertex of a non-trivial block in $G$. We show the existence of a pair $y^e, z^e \in L^e \setminus \{x^e\}$ in $V^J$ such that $TIA(\{x^e, y^e, \alpha_1\}, \{x^e, z^e, \alpha_2^e\})$ returns FALSE, where $\alpha_1^e, \alpha_2^e \in V^J \setminus L^e$. From Section 3 we have that any non-trivial block in $G$ has at least three vertices. Hence, there exists another vertex $y^e \in L^e$ for which the noiseless counterpart is a non-cut vertex. We will show that one of $\{x^e, y^e, \alpha_1^e\}$ and $\{x^e, z^e, \alpha_2^e\}$ is not a star triplet, where $z^e \in L^e \setminus x^e, y^e$, and $\alpha_1^e, \alpha_2^e \in V^J \setminus L^e$. Then, the $TIA(\{x^e, y^e, \alpha_1^e\}, \{x^e, z^e, \alpha_2^e\})$ being FALSE will follow from Lemma B.7. As $x$ and $y$ both are non-cut vertices, there does not exist a cut vertex that separates $x$ and $y$ in $G$, which implies that there does not exist an ancestor $a$ in $G^J$ s.t. $x^e \perp\!\!\!\perp y^e | a$. Hence, $\{x^e, y^e, \alpha_1^e\}$ is not a star triplet in $G^J$. Then, the proof follows from Lemma B.7.

($\Leftarrow$) Notice that the pair $(\{x^e, y^e, \alpha_1^e\}, \{x^e, z^e, \alpha_2^e\})$ can not share an ancestor, as it would violate the claim that $\{x^e, y^e, z^e\}$ is in a leaf cluster. Then, from Lemma B.7 we have that if $TIA(\{x^e, y^e, \alpha_1^e\}, \{x^e, z^e, \alpha_2^e\})$ returns FALSE, then at least one of the triplets is a non-star triplet, which rules out the existence of star triplets with non-identical ancestor. Suppose that $\{x^e, y^e, \alpha_1^e\}$ is a non-star triplet. As $\alpha_1^e \notin L^e$, an ancestor separates $x^e$ and $\alpha_1^e$, and $y^e$ and $\alpha_1^e$. Then, the ancestor identified above does not separate $x^e$ and $y^e$. Hence in $G$, there does not exist a cut vertex that separates $x$ and $y$. □

Unidentifiability of the articulation point from a leaf cluster. According to Lemma B.12 the NONCUTTEST returns the non-cut vertices of a non-trivial block from a leaf cluster, and the next (immediate) step is to learn the cut vertices of the non-trivial blocks. We now present a claim which shows a case where identifying the articulation point from a leaf cluster is not possible. This ambiguity is exactly the ambiguity (in robust model selection problem) of the *label swapping of the leaf vertices with their neighboring internal vertices* of a tree-structured Gaussian graphical models Katiyar et al. (2019).

**Claim 5.** *Let (i) a vertex $v \in V_{\mathrm{cut}}$ separates a subset $L \subset 2^V$ of vertices from any $v' \in V_{\mathrm{cut}} \setminus v$ where $L$ contains at least one leaf vertex, and (ii) $L^e$ be the noisy counterpart of $L$. Then, there exist at least two vertices $x_1^e, x_2^e \in L^e \cup \{v^e\}$ such that $TIA(\{x^e, y^e, \alpha_1^e\}, \{x^e, z^e, \alpha_2^e\})$ returns TRUE for any pair $y^e, z^e \in L^e \cup \{v^e\}$ where $x^e \in \{x_1^e, x_2^e\}$, and $\alpha_1^e, \alpha_2^e \in V^{\mathrm{J}} \setminus L^e \cup \{v^e\}$.*

*Proof.* As $v$ is a cut vertex, $v^e \perp\!\!\!\perp y^e | v$, $v^e \perp\!\!\!\perp \alpha_1^e | v$, and $y^e \perp\!\!\!\perp \alpha_1^e | v$. Here, $v$ is a unique separator since no other cut vertex (or ancestor in $G^{\mathrm{J}}$) separates $v^e$ and $y^e$. Hence, $v$ is the ancestor of $\{v^e, y^e, \alpha_1^e\}$ in $G^{\mathrm{J}}$. Similarly, one can construct another triplet $\{v^e, x^e, \alpha_1^e\}$ which has an ancestor $v$ in $G^{\mathrm{J}}$. Hence, $TIA(\{v^e, y^e, \alpha_1^e\}, \{v^e, x^e, \alpha_2^e\})$ will return TRUE. Now, let us consider a leaf vertex $x_1$ in $L$. Now, $x_1^e \perp\!\!\!\perp \alpha_1^e | v$, $x_1^e \perp\!\!\!\perp y^e | v$, and $\alpha_1^e \perp\!\!\!\perp y^e | v$. Hence, $v$ is the ancestor of $\{x_1^e, y^e, \alpha_1^e\}$. Similarly, one can construct another triplet such that $v$ is the ancestor of $\{x_1^e, x^e, \alpha_2^e\}$. Hence, $TIA(\{x_1^e, y^e, \alpha_1^e\}, \{x_1^e, x^e, \alpha_2^e\})$ will return TRUE. $\square$

**Proposition B.13.** *Suppose that Subroutine 6 is invoked with the correct leaf clusters and internal clusters. Further suppose that NONCUTTEST succeeds in identifying the non-cut vertices of a non-trivial block. Then, Subroutine 6 correctly learns $\mathcal{P}_{\mathrm{op}}$ and $A_{\mathrm{op}}$ for $\mathcal{T}_{\mathrm{op}}$.*

*Proof.* According to Lemma B.12, Subroutine 6 correctly learns the non-cut vertices of any non-trivial block $I$ with more than one cut vertices. If the cut vertex is observed, then it is identified in Subroutine 3, and declared as one the articulation points of the vertex $I$ in $\mathcal{P}_{\mathrm{op}}$. Otherwise, the noisy counterpart belongs to a leaf cluster associated with an hidden ancestor, and the cut vertex be identified by selecting the label of the leaf cluster which is associated with the hidden ancestor (unobserved cut vertex of non-trivial block.)

$\square$

We now establish the correctness of NoMAD in the learning the edge set $E_{\mathrm{op}}$ for $\mathcal{T}_{\mathrm{op}}$. This goal is achieved correctly by Procedure NONBLOCKNEIGHBORS of NoMAD.

**Proposition B.14.** *Suppose that Procedure 4 is invoked with the correct $\mathcal{P}_{\mathrm{op}}$ and $A_{\mathrm{op}}$. Then, Procedure 4 returns the edge set $E_{\mathrm{op}}$ correctly.*

*Proof.* Procedure NONBLOCKNEIGHBORS correctly learns the neighbors of any fixed articulation point in $A_{\mathrm{op}}$ by ruling out the non-neighbor articulation points in $\mathcal{T}_{\mathrm{op}}$. First, the procedure gets rid of the articulation points which are separated from the articulation points of the same vertex in $A_{\mathrm{op}}$. Then, from the remaining articulation points it chooses the set of all those articulation points such that no pair in the set is separated from each other by the fixed articulation point. Then, Procedure 4 creates edges between vertices which contains the neighboring articulation points. $\square$

**Constructing the Equivalence Class.** Finally, in order to show that we can construct the equivalence class $[G]$ from the articulated set tree $\mathcal{T}_{\mathrm{op}}$, we note some additional definitions in the following. For graph $G$, let $B_{\mathrm{non\text{-}cut}}$ be the set of all non-cut vertices in a non-trivial block $B$. Define $\mathcal{B}_{\mathrm{non\text{-}cut}} \triangleq \bigcup_{B \in \mathcal{B}^{\mathrm{NT}}} B_{\mathrm{non\text{-}cut}}$, where $\mathcal{B}^{\mathrm{NT}}$ is the set of all non-trivial blocks. Let a set $F_i$ referred as a *family* be defined as $\{v : \deg(v) = 1 \text{ and } \{v, i\} \in E(G)\} \cup \{i\}$ where $E(G)$ is the edge set of $G$, and let $\mathcal{F} = \bigcup_{i \in V} F_i$. Let $K$ be the set of cut vertices whose neighbors do not contain a leaf vertex in $G$. For any vertex $k \in K$, let a family $F_k \in \mathcal{F}$ be such that there exists a vertex $f \in F_k$ such that $\{k, f\} \in E(G)$. For example, in Fig. 1a, $\mathcal{F} = \{\{10, 11, 12, 13\}, \{14, 15, 16\}, \{17, 20, 21\}\}$; two sets $\{1, 2, 3\}, \{18, 19\}$ in $\mathcal{B}_{\mathrm{non\text{-}cut}}$, and $K = \{4, 6, 7, 8, 9\}$. Also, for example, $F_4 = \{10, 11, 12, 13\}$. For any arbitrary graph $\widetilde{G}$, let $B_{\mathrm{non\text{-}cut}}(\widetilde{G})$, $\mathcal{F}(\widetilde{G})$, and $K(\widetilde{G})$ be the corresponding sets from $\widetilde{G}$.

Now, notice that in $\mathcal{T}_{\mathrm{op}}$, each vertex $k \in K$ has at least an edge in $\mathcal{T}_{\mathrm{op}}$. Let $N_{\mathrm{art}}(k)$ be the neighbors of $k \in K$ in the edge set $E_{\mathrm{op}}$ returned for $\mathcal{T}_{\mathrm{op}}$. Now, notice that as as long as $\mathcal{B}_{\mathrm{non\text{-}cut}}$, $\mathcal{F}$, and $K$ are identified correctly in $\mathcal{T}_{\mathrm{op}}$, and the following condition holds in $E_{\mathrm{op}}$ for any $i \in N_{\mathrm{art}}(k)$ for each $k \in K$: (a) if $i \in K$, then $\{i, k\} \in E(G)$, and (b) otherwise, there exists a vertex $j \in F_k$ such that $\{j, k\} \in E(G)$. Informally, identifying $\mathcal{B}_{\mathrm{non\text{-}cut}}$ and $\mathcal{F}$ correctly, makes sure that vertices that constructs the local neighborhoods of any graph in $[G]$ are identical; identifying $K$ correctly, and satisfying the above-mentioned condition makes sure that the correct articulation points are recovered. Notice that the sets $\mathcal{B}_{\mathrm{non\text{-}cut}}$, $\mathcal{F}$, and $K$ are identical in all

the graphs in Fig. 2. Following proposition shows that the sets $\mathcal{B}_{\text{non-cut}}$, $\mathcal{F}$, and $K$ are identified correctly from $\mathcal{T}_{\text{op}}$.

**Lemma B.15.** *Let $\widetilde{G}$ be an arbitrary graph. Then, $\widetilde{G} \in [G]$ if and only if the following holds:*

1. $\mathcal{B}_{\text{non-cut}}(\widetilde{G}) = \mathcal{B}_{\text{non-cut}}(G)$, $\mathcal{F}(\widetilde{G}) = \mathcal{F}(G)$, *and* $K(\widetilde{G}) = K(G)$.

2. *For any vertex $k \in K$, let a family $F_k \in \mathcal{F}$ be such that there exists a vertex $f \in F_k$ such that $\{k, f\} \in E(\widetilde{G})$. Now, for any neighbor $i \in N(k)$: (a) if $i \in K$, then $\{i, k\} \in E(\widetilde{G})$, and (b) otherwise, there exists a vertex $j \in F_k$ such that $\{j, k\} \in E(\widetilde{G})$.*

*Proof.* ($\Rightarrow$) The forward implication follows from Definition 3.2.

($\Leftarrow$) For the reverse implication, notice that the first condition is associated with the equality between sets. $\mathcal{B}_{\text{non-cut}}(\widetilde{G}) = \mathcal{B}_{\text{non-cut}}(G)$ implies non-cut vertices are identified correctly, and $\mathcal{F}(\widetilde{G}) = \mathcal{F}(G)$ implies families are identified correctly. The second condition implies that an edge associated with a vertex $k \in K$ will have an ambiguity when the other vertex is from a family. Recall that from Definition 3.2, the label of a cut vertex can be swapped with it's neighbor leaf vertices. $\qquad\square$

The reverse implication of the above-mentioned proof can be understood as follows: Identifying $\mathcal{B}_{\text{non-cut}}$ and $\mathcal{F}$ ensures that essentially the *local structures* are identical between $G$ and $\widetilde{G}$. Recovering $K$ correctly and satisfying the second condition ensure that these local structures are correctly attached at the appropriate points.

**Proposition B.16** (Correctness in Learning the Equivalence Class). *Suppose that $\mathcal{P}_{\text{op}}$, $A_{\text{op}}$, and $E_{\text{op}}$ returned by $\mathcal{T}_{\text{op}}$ is correct. Then, following is true: (a)The sets $\mathcal{B}_{\text{non-cut}}$, $\mathcal{F}$, and $K$ are identified correctly, and (b) the condition is true for $N_{\text{art}}(k)$ for each $k \in K$.*

*Proof.* We first show that NoMAD correctly identifies the sets $\mathcal{B}_{\text{non-cut}}$, $\mathcal{F}$, and $K$. By Lemma B.12, Subroutine 6 correctly identifies the set $\mathcal{B}_{\text{non-cut}}$. Now, recall that each $F \in \mathcal{F}$ is a set of vertices constructed with a cut vertex and its neighbor leaf vertices. Hence, each family $F \in \mathcal{F}$ is captured in one of the leaf clusters returned by Subroutine 4. As Subroutine 6 correctly identifies the non-cut vertices from each leaf cluster, $\mathcal{F}$ is identified correctly. Finally, by Claim 5, the ambiguity in learning an articulation point is present only when a cut vertex has leaf vertex as it's neighbor; but $K$ does not contain such cut vertices. Hence, Subroutine 6 correctly learns $K$. We now show that above-mentioned condition is satisfied for the neighbor articulation points in $N_{\text{art}}(k)$ for any $k \in K$. As $K$ are identified correctly by Subroutine 6, and the Procedure 4 returns correct $N_{\text{art}}(k)$, it is clear that if any neighbor articulation point $i \in N_{\text{art}}(k) \cap K$, then $\{i, k\} \in E(G)$. Now, suppose that $i \notin N_{\text{art}}(k) \cap K$. Then, from Definition 3.2, the label of a cut vertex can be swapped with it's neighbor leaf vertices. As each family $F \in \mathcal{F}$ are identified correctly, there exists a vertex $j \in F_k$ (which is an *unidentified* cut vertex in $G$) such that $\{i, j\} \in E(G)$. $\qquad\square$

## C Sample Complexity Result

Recall that NoMAD returns the equivalence class of a graph $G$ while having access only to the noisy samples according to the problem setup in Section 3.1. But, in the finite sample regime, instead of the population quantities, we only have access to samples. We will use these to create natural estimates $\widehat{\rho}_{ij}$, for all $i, j \in V^o$ of the correlation coefficients given by $\widehat{\rho}_{ij} \triangleq \frac{\widehat{\Sigma^o}_{ij}}{\sqrt{\widehat{\Sigma^o}_{ii}\widehat{\Sigma^o}_{jj}}}$, where $\widehat{\Sigma^o}_{ij} = \frac{1}{n}\sum_{k=1}^{n} y_i^{(k)} y_j^{(k)}$. Indeed, these are random quantities and therefore we need to make slight modifications to the algorithm as follows:

**Change in the TIA test.** We start with the following assumption: For any triplet pair $U, W \in \binom{V}{3} \setminus \mathcal{V}_{\text{star}} \cup \mathcal{V}_{\text{sep}}$ and any vertex pair $(x, a) \in U \times W$, there exists a constant $\zeta > 0$, such that $\left|d_x^U + d_a^W - d_{xa}\right| > \zeta$. As we showed in Lemma B.7, for any pair $U, W \in \binom{V}{3} \setminus \mathcal{V}_{\text{star}} \cup \mathcal{V}_{\text{sep}}$, there exists at least one triplet $\{x, a, b\}$ where $x \in U$ and $a, b \in W$ such that $d_{xa} - d_x^U - d_a^W \neq 0$ and $d_{xb} - d_x^U - d_b^W \neq 0$. Hence, the observation

in Lemma B.7 motivates us to replace the exact equality testing in the TIA test in Definition 3.5 with the following hypothesis test against zero: $\max\left\{\left|\widehat{d}_{xa} - \widehat{d}_x^U - \widehat{d}_a^W\right|, \left|\widehat{d}_{xb} - \widehat{d}_x^U - \widehat{d}_b^W\right|\right\} \le \xi$, for some $\xi < \frac{\zeta}{2}$.

**Change in the Mode test.** In order to compute the distance between the hidden ancestors in the finite sample regime, we first recall from (the proof of) Lemma B.9 that there are at least 4 instances (w.l.o.g.) $\Delta(x_1, y_1), \Delta(x_1, y_2), \Delta(x_2, y_1)$, and $\Delta(x_2, y_2)$ where $\Delta(x, y)$ where $x \in U_i$ and $y \in U_j$ such that equals to $d_{ij}$. We also showed that no set of identical but incorrect distance has cardnilaity more than two. Hence, In the finite sample regime, we replace the mode test in Subroutine 3 with a more robust version, which we call the $\epsilon_d$ − mode test, where $\epsilon_d < \min(\frac{\xi}{14}, \gamma)$ based on the following definition.

**Definition C.1** ($\epsilon_d$ − mode). *Given a set of real numbers $\{r_1, \ldots, r_n\}$, let $S_1, \ldots, S_k$ be a partition where each $r, r' \in S_i$ is such that $|r - r'| < \epsilon_d$ for each $i$. Then, the $\epsilon_d$-mode of the this set is defined as selecting an arbitrary number from the partition with the largest cardinality.*

In the finite sample regime, we run NoMAD with the mode replaced by the $\epsilon_d$-mode defined above such that $\epsilon_d < \min(\frac{\xi}{14}, \gamma)$. We will call this modified mode test as the $\epsilon_d$-mode test.

**Change in Separation test.** For any triplet $(i, j, k) \in \binom{V^o}{3}$, in order to check whether $i \perp\!\!\!\perp j|k$, instead of the equality test in Fact 1, we modified the test for the finite sample regime as follows: $|\widehat{d}_{ij} - \widehat{d}_{ik} - \widehat{d}_{jk}| < \frac{\epsilon_d}{6}$. We now introduce two new notations to state our main result. Let $\rho_{\min}(p) = \min_{i,j \in \binom{p}{2}}|\rho_{ij}|$ and $\kappa(p) = \log((16 + (\rho_{\min}(p))^2 \epsilon_d^2)/(16 - (\rho_{\min}(p))^2 \epsilon_d^2))$, where $\epsilon_d = \min(\frac{\xi}{14}, \gamma)$, where $\gamma$ is from Assumption 5.2.

**Theorem C.3.** *Suppose the underlying graph $G$ of a faithful GGM satisfies Assumptions 5.2-5.3. Fix any $\tau \in (0, 1]$. Then, there exists a constant $C > 0$ such that if the number of samples $n$ satisfies $n > C\left(\frac{1}{\kappa(p)}\right) \max\left(\log\left(\frac{p^2}{\tau}\right), \log\left(\frac{1}{\kappa(p)}\right)\right)$, then with probability at least $1 - \tau$, NoMAD accepting $\widehat{d}_{ij}$ outputs the equivalence class $[G]$.*

*Proof.* First, there are (at most) seven pairwise distances to be estimated in terms of $\max\left\{\left|\widehat{d}_{xa} - \widehat{d}_x^U - \widehat{d}_a^W\right|, \left|\widehat{d}_{xb} - \widehat{d}_x^U - \widehat{d}_b^W\right|\right\}$. Therefore, the probability that our algorithm fails is bounded above by the probability that there exists a pairwise distance estimate that is $\xi/14$ away from its mean. To this end, let us denote a bad event $B_{i,j}$ for any pair $i, j \in V^O$ as the following:

$$B_{i,j} \triangleq \{|d_{ij} - \widehat{d}_{ij}| \ge \epsilon_d\}. \tag{6}$$

Then, the error probability $\mathbb{P}[[\mathcal{T}_{\text{algo}}] \ne [G]]$ is upper bounded as

$$\mathbb{P}([\mathcal{T}_{\text{algo}}] \ne [G]) \le \mathbb{P}\left(\bigcup_{i,j \in V^o} B_{i,j}\right) \le \sum_{i,j \in V^o} \mathbb{P}(B_{i,j}), \tag{7}$$

where $[\mathcal{T}_{\text{algo}}]$ is the output equivalence class. We now consider two following events: $K_{i,j} \triangleq \{|\widehat{\rho}_{ij}| \le \frac{\rho_{\min}}{2}\}$ [6], and $R_{i,j} \triangleq \{|\rho_{ij} - \widehat{\rho}_{ij}| < \frac{\rho_{\min}\epsilon_d}{2}\}$. We will upper bound $\mathbb{P}(B_{i,j})$ for any pair $i, j$ using $\mathbb{P}(K_{i,j})$ and $\mathbb{P}(R_{i,j})$. Before that, notice the following chain of implications:

$\left(|\rho_{ij} - \widehat{\rho}_{ij}| < \frac{\rho_{\min} \times \epsilon_d}{2}\right) \Rightarrow \left(||\rho_{ij}| - |\widehat{\rho}_{ij}|| < \frac{\rho_{\min} \times \epsilon_d}{2}\right) \Rightarrow \left(\left|d_{ij} - \widehat{d}_{ij}\right| < \frac{||\rho_{ij}| - |\widehat{\rho}_{ij}||}{\min(|\widehat{\rho}_{ij}|, |\rho_{ij}|)}\right) \Rightarrow$

$\left(\left|d_{ij} - \widehat{d}_{ij}\right| < \frac{||\rho_{ij}| - |\widehat{\rho}_{ij}||}{\min(\frac{\rho_{\min}}{2}, \rho_{\min})}\right) \Rightarrow \left(\left|d_{ij} - \widehat{d}_{ij}\right| < \frac{\frac{\rho_{\min}}{2} \times \epsilon_d}{\frac{\rho_{\min}}{2}}\right) \Rightarrow \left(\left|d_{ij} - \widehat{d}_{ij}\right| < \epsilon_d\right)$. These implications establish that $R_{i,j} \cap K_{i,j}^c \subseteq B_{i,j}^c$. Notice that as $R_{i,j} \cap K_{i,j}^c \subseteq B_{i,j}^c \cap K_{i,j}^c$, it will imply that $\mathbb{P}(B_{i,j}^c \cap K_{i,j}^c) \ge \mathbb{P}(R_{i,j} \cap K_{i,j}^c)$. Now, we can write the following bound:

$$\mathbb{P}(B_{i,j}|K_{i,j}^c) \le \mathbb{P}(R_{i,j}^c|K_{i,j}^c). \tag{8}$$

---

[6]for notational clarity we write $\rho_{\min}$ instead of $\rho_{\min}(p)$

Then, $\mathbb{P}(B_{ij})$ can be upper bounded as follows:

$$\mathbb{P}\left(B_{i,j}\right) = \mathbb{P}\left(B_{i,j}|K_{i,j}\right)\mathbb{P}\left(K_{i,j}\right) + \mathbb{P}\left(B_{i,j}|K_{i,j}^c\right)\mathbb{P}\left(K_{i,j}^c\right), \tag{9}$$

$$\leq \mathbb{P}\left(B_{i,j}|K_{i,j}\right)\mathbb{P}\left(K_{i,j}\right) + \mathbb{P}\left(R_{i,j}^c|K_{i,j}^c\right)\mathbb{P}\left(K_{i,j}^c\right), \tag{10}$$

$$\leq \left(1 \times \mathbb{P}\left(K_{i,j}\right)\right) + \left(\mathbb{P}\left(R_{i,j}^c|K_{i,j}^c\right) \times 1\right). \tag{11}$$

Then, $\mathbb{P}\left([\mathcal{T}_{\text{algo}}] \neq [G]\right)$ can be further bounded as

$$\mathbb{P}\left([\mathcal{T}_{\text{algo}}] \neq [G]\right) \leq \sum_{i,j \in V^{\text{o}}} \mathbb{P}\left(B_{i,j}\right) \leq \sum_{i,j \in V^{\text{o}}} \mathbb{P}\left(K_{i,j}\right) + \sum_{i,j \in V^{\text{o}}} \mathbb{P}\left(R_{i,j}^c|K_{i,j}^c\right).$$

Because $\mathbb{P}\left(R_{i,j}^c|K_{i,j}^c\right) < \mathbb{P}\left(R_{i,j}^c\right)/\mathbb{P}\left(K_{i,j}^c\right)$, we note that

$$\mathbb{P}\left([\mathcal{T}_{\text{algo}}] \neq [G]\right) \leq \sum_{i,j \in V^{\text{o}}} \mathbb{P}\left(K_{i,j}\right) + \sum_{i,j \in V^{\text{o}}} \frac{\mathbb{P}\left(R_{i,j}^c\right)}{\mathbb{P}\left(K_{i,j}^c\right)}.$$

We now find the required number of samples $n$ in order for $\mathbb{P}\left([\mathcal{T}_{\text{algo}}] \neq [G]\right)$ to be bounded by $\tau$. Before computing $n$ we note an important inequality from Kalisch & Bühlman (2007) which we use in bounding all the following events. For any $0 < \epsilon \leq 2$, and $\sup_{i \neq j} |\rho_{ij}| \leq M < 1$, following is true.

$$\mathbb{P}\left(|\widehat{\rho}_{ij} - \rho_{ij}| > \epsilon\right) \leq C_\rho \left(n-2\right) \exp\left(-\left(n-4\right)\log\left(\frac{4+\epsilon^2}{4-\epsilon^2}\right)\right), \tag{12}$$

for some constant $0 < C_\rho < \infty$ depending on $M$ only.

We now note the following assumption on bounded correlation which is a common assumption in learning the graphical models: $0 < \rho_{\min} \leq \rho_{\max} < 1$. Now notice that, $\left(|\widehat{\rho}_{ij}| \leq \frac{\rho_{\min}}{2}\right)$ together with $|\rho_{ij}| \geq \rho_{\min}$ implies that $|\rho_{ij}| - |\widehat{\rho}_{ij}| \geq \rho_{\min} - \frac{\rho_{\min}}{2} = \frac{\rho_{\min}}{2}$, since $\rho_{\min} > \frac{\rho_{\min}}{2}$. Furthermore, $|\rho_{ij} - \widehat{\rho}_{ij}| \geq |\rho_{ij}| - |\widehat{\rho}_{ij}|$ implies that $|\rho_{ij} - \widehat{\rho}_{ij}| \geq \frac{\rho_{\min}}{2}$. Then, we have the following:

$$\mathbb{P}\left(K_{i,j}\right) \leq \mathbb{P}\left(|\rho_{ij} - \widehat{\rho}_{ij}| \geq \frac{\rho_{\min}}{2}\right) \leq C_\rho \left(n-2\right) \exp\left(-\left(n-4\right)\log\left(\frac{16+\rho_{\min}^2}{16-\rho_{\min}^2}\right)\right). \tag{13}$$

Eq. equation 13 follows from Eq. equation 12. Now, According to Claim 6.

$$n_1 > \max\left(C_1 \frac{\log\left(\frac{2C_\rho\binom{p}{2}}{\tau}\right)}{\log\left(\frac{16+\rho_{\min}^2}{16-\rho_{\min}^2}\right)} \times \frac{C_2 C_1}{(C_1-1)\log\left(\frac{16+\rho_{\min}^2}{16-\rho_{\min}^2}\right)}, \log\left(\frac{C_1}{(C_1-1)\log\left(\frac{16+\rho_{\min}^2}{16-\rho_{\min}^2}\right)}\right)\right) + 4 \tag{14}$$

implies $\sum\limits_{i,j \in V^{\text{o}}} \mathbb{P}(K_{i,j}) < \frac{\tau}{2}$,

$$n_3 > \max\left(C_1 \frac{\log\left(\frac{C_\rho}{1-\tau'}\right)}{\log\left(\frac{16+\rho_{\min}^2}{16-\rho_{\min}^2}\right)}, \frac{C_2 C_1}{(C_1-1)\log\left(\frac{16+\rho_{\min}^2}{16-\rho_{\min}^2}\right)} \times \log\left(\frac{C_1}{(C_1-1)\log\left(\frac{16+\rho_{\min}^2}{16-\rho_{\min}^2}\right)}\right)\right) + 4 \tag{15}$$

implies $\mathbb{P}(K_{i,j}^c) > \tau'$, where $\tau' > 1 - C_\rho$, and

$$n_4 > \max\left(C_1 \frac{\log\left(\frac{2C_\rho\binom{p}{2}}{\tau\tau'}\right)}{\log\left(\frac{16+\rho_{\min}^2\epsilon_{\text{d}}^2}{16-\rho_{\min}^2\epsilon_{\text{d}}^2}\right)}, \frac{C_2 C_1}{(C_1-1)\log\left(\frac{16+\rho_{\min}^2\epsilon_{\text{d}}^2}{16-\rho_{\min}^2\epsilon_{\text{d}}^2}\right)} \times \log\left(\frac{C_1}{(C_1-1)\log\left(\frac{16+\rho_{\min}^2\epsilon_{\text{d}}^2}{16-\rho_{\min}^2\epsilon_{\text{d}}^2}\right)}\right)\right) + 4 \tag{16}$$

implies $\mathbb{P}(R_{i,j}^c) < \frac{\tau\tau'}{2\binom{p}{2}}$. Now, notice that $n_2 \triangleq \max(n_3, n_4)$ implies $\frac{\mathbb{P}(R_{i,j}^c)}{\mathbb{P}(K_{i,j}^c)} < \frac{\tau}{2\binom{p}{2}}$. Therefore, acquiring at least $n_2$ samples will imply $\sum_{i,j \in V^o} \frac{\mathbb{P}(R_{i,j}^c)}{\mathbb{P}(K_{i,j}^c)} < \frac{\tau}{2}$. Finally, for $\mathbb{P}([\mathcal{T}_{\text{algo}}] \neq [G])$ to be upper bounded by $\tau$, it is sufficient for the number of samples $n$ to satisfy $n > \max(n_1, n_2)$. $\qquad\square$

**Claim 6.** *There exist positive constants $T, C,$ and $\widetilde{\alpha}$ such that if $n > \max(T, C \times \widetilde{\alpha} \log \widetilde{\alpha})$, then $n - \widetilde{\alpha} \log(n) > T$.*

*Proof.* We start the proof with the following claim: Suppose that there exists a constant $C_1, C_2$ where $C_1 < C_2$ such that $C_1 m \log m < n < C_2 m \log m$. Notice that for $m$ sufficiently large ($m > C_2$), we can show that $n > m \log n$. Therefore, for some constant $C_1, C_2$, $n > C_2 \times \frac{C_1}{(C_1-1)\alpha} \log\left(\frac{C_1}{(C_1-1)\alpha}\right)$ implies $n > \frac{C_1}{(C_1-1)\alpha} \log(n)$. Now, suppose that $\max\left(C_1 T, \frac{C_2 C_1}{(C_1-\alpha)} \log\left(\frac{C_1}{(C_1-1)\alpha}\right)\right) = C_1 T$. Then, $n > C_1 T$ implies $n > C_2 \times \frac{C_1}{(C_1-1)\alpha} \log\left(\frac{C_1}{(C_1-1)\alpha}\right)$. Then, from the initial claim we have that $n > \frac{C_1}{(C_1-1)\alpha} \log(n)$. Then, $n \frac{(C_1-1)}{C_1} > \frac{1}{\alpha} \log(n)$, and $n - \frac{1}{\alpha} \log(n) > \frac{n}{C_1}$. As $\frac{n}{C_1} > T$, we have that $n - \frac{1}{\alpha} \log(n) > T$. Further, suppose that $\max\left(C_1 T, \frac{C_2 C_1}{(C_1-\alpha)} \log\left(\frac{C_1}{(C_1-1)\alpha}\right)\right) = \frac{C_2 C_1}{(C_1-\alpha)} \log\left(\frac{C_1}{(C_1-1)\alpha}\right)$. Then, from the initial claim we have that $n > \frac{C_2 C_1}{(C_1-\alpha)} \log\left(\frac{C_1}{(C_1-1)\alpha}\right)$ implies $n > \frac{C_1}{(C_1-1)\alpha} \log(n)$. Also, $n > \frac{C_2 C_1}{(C_1-\alpha)} \log\left(\frac{C_1}{(C_1-1)\alpha}\right)$ implies $n > C_1 T$, which will imply $n - \frac{1}{\alpha} \log(n) > \frac{n}{C_1} > T$. Setting $\widetilde{\alpha}$ equals to $\frac{C_1}{(C_1-1)\alpha}$ proves the result. $\qquad\square$

# D  Identifiability Result

*Proof.* We first consider the case where there is only one non-trivial block $\mathcal{B}^{NT}$ inside $G$ and that the block cut vertices of $\mathcal{B}^{NT}$ do not have neighboring leaf nodes. As a result, $\mathcal{B}^{NT}$ contains exactly two block cut vertices $b_1$ and $b_2$ connected to the cut vertices $p_1$ and $p_2$, respectively. Thus, we express the vertex set $V$ of $G$ as a union of disjoint sets $V_1 \cup \{p_1\}$, $V_2 \cup \{p_2\}$, and $V_{NT}$—the vertex set of $\mathcal{B}^{NT}$.

Without loss of generality, let $V_1 \cup \{p_1\} = \{1, \ldots, p_1\}$, $V_{NT} = \{p_1 + 1, \ldots, p_2 - 1\}$, and $V_2 \cup \{p_2\} = \{p_2, \ldots, p\}$. Also, let $b_1 = p_1 + 1$ and $b_2 = p_2 - 1$. Because $G$, it follows that $V_1 \cup \{p_1\} \perp\!\!\!\perp V_1 \cup \{p_1\} \mid V_{NT}$. In words, $V_{NT}$ separates $V_1 \cup \{p_1\}$ and $V_2 \cup \{p_2\}$. Furthermore, $b_1$ shares an edge with $p_1$ and $b_2$ shares an edge with $p_2$. From these facts, $K^* = (\Sigma^*)^{-1}$ can be partitioned as in equation 17 (see below). Let $K_1$, $K_{NT}$, and $K_2$ be the first, second, and third diagonal blocks of $K^*$ in equation 17. Let $e_j$ be the canonical basis vector in $\mathbb{R}^p$. Then, we can express $K^*$ in equation 17 as

$$K^* = \text{Blkdiag}(K_1, K_{NT}, K_2) + e_{p_1+1} e_{p_1}^\mathsf{T} K_{p_1+1,p_1} + e_{p_1} e_{p_1+1}^\mathsf{T} K_{p_1,p_1+1} + e_{p_2-1} e_{p_2}^\mathsf{T} K_{p_2-1,p_2} + e_{p_2} e_{p_2-1}^\mathsf{T} K_{p_2,p_2-1}. \quad (18)$$

Recall that $\Sigma^0 = \Sigma^* + D$. Decompose the diagonal matrix $D$ as $D = D^{(1)} + D^{(2)}$, where

$$D^{(1)} = \text{Blkdiag}(\mathbf{0}, D_{NT}^{(1)}, \mathbf{0}), \quad (19)$$

$$D^{(2)} = \text{Blkdiag}(D_1, D_{NT}^{(2)}, D_2), \quad (20)$$

$$K^* = \left[\begin{array}{ccc|ccc|ccc}
K_{11} & \ldots & K_{1,p_1} & 0 & \ldots & 0 & & & \\
\vdots & \ddots & \vdots & \vdots & \ddots & \vdots & & & \\
K_{p_1,1} & \ldots & K_{p_1,p_1} & K_{p_1+1,p_1} & \ldots & 0 & & & \\
\hline
0 & \ldots & K_{p_1,p_1+1} & K_{p_1+1,p_1+1} & \ldots & K_{p_1+1,p_2-1} & 0 & \ldots & 0 \\
\vdots & \ddots & \vdots & \vdots & \ddots & \vdots & \vdots & \ddots & \vdots \\
0 & \ldots & 0 & K_{p_2-1,p_1+1} & \ldots & K_{p_2-1,p_2-1} & K_{p_2-1,p_2} & \ldots & 0 \\
\hline
& & & 0 & \ldots & K_{p_2,p_2-1} & K_{p_2,p_2} & \ldots & K_{p_2,p} \\
& & & \vdots & \ddots & \vdots & \vdots & \ddots & \vdots \\
& & & 0 & \ldots & 0 & K_{p,p_2} & \ldots & K_{p,p}
\end{array}\right] \quad (17)$$

and the dimensions of $D_1$, $D_{NT}$, and $D_2$ are same as those of $K_1$, $K_{NT}$, and $K_2$, resp. Furthermore, $D_{NT}^{(1)} = \mathrm{diag}(0, \times, \ldots, \times, 0)$ and $D_{NT}^{(2)} = \mathrm{diag}(\times, 0, \ldots, 0, \times)$. Here $\times$ can be a zero or a positive value. Let $\Sigma^q = \Sigma^* + D^{(1)}$ and $D^q = D^{(2)}$. From the above notations, we have $\Sigma^0 = \Sigma^* + D = \Sigma^* + D^{(1)} + D^{(2)} = \Sigma^q + D^q$. We show that there exists a decomposition of $D$ into $D_1$ and $D_2$ such that the inverse of $\Sigma^q \triangleq \Sigma^* + D^{(1)}$ has different structure. It suffices to show that $(\Sigma^q)^{-1}$ exactly equals the expression of $K^*$ in equation 18, except for the second diagonal block $K_{NT}$ in $\mathrm{Blkdiag}(K_1, K_{NT}, K_2)$. Recall that different values of $K_{NT}$ yield different subgraphs on the non-trivial block, and consequently, different graphs in $[G]$; see Definition 3.1. Consider the following identity:

$$(\Sigma^q)^{-1} = (\Sigma^* + D^{(1)})^{-1} = (I + (\Sigma^*)^{-1} D^{(1)})^{-1} (\Sigma^*)^{-1} = (I + K^* D^{(1)})^{-1} K^*. \tag{21}$$

We first evaluate $(I + K^* D^{(1)})^{-1}$. Note that $e_{p_1+1}$, $e_{p_1}$, $e_{p_2-1}$, and $e_{p_2}$ lie in the nullspace of $D^{(1)}$ and $K^* D^{(1)}$. Using this fact and the formulas in equation 18 and equation 19, we can simplify $(I + K^* D^{(1)})$ as

$$(I + K^* D^{(1)}) = \mathrm{Blkdiag}(I, I + K_{NT} D_{NT}^{(1)}, I), \tag{22}$$

where, $\widetilde{K}_{NT} \triangleq I + K_{NT} D_{NT}^{(1)}$ is a positive definite matrix, and hence, invertible. This is because $K_{NT} D_{NT}^{(1)}$ and $(D_{NT}^{(1)})^{1/2} K_{NT}^{1/2} K_{NT}^{1/2} (D_{NT}^{(1)})^{1/2}$ are similar matrices, where we used the facts that $K_{NT}$ is positive definite and $D_{NT}^{(1)}$ is non-negative diagonal. Thus,

$$(I + K^* D^{(1)})^{-1} = \mathrm{Blkdiag}(I_{p_1}, \widetilde{K}_{NT}^{-1}, I_{p-p_2+1}). \tag{23}$$

Also, note that the null space vectors $e_{p_1+1}$, $e_{p_1}$, $e_{p_2-1}$, and $e_{p_2}$ of $K^* D^{(1)}$ are also the eigenvectors of $(I + K^* D^{(1)})^{-1}$, with eigenvalues all being equal to one. Putting together the pieces, from equation 18, equation 21, and equation 23 we have $(\Sigma^q)^{-1} = (I + K^* D^{(1)})^{-1} K^*$ which equals to the following:

$$= \mathrm{Blkdiag}(K_1, \widetilde{K}_{NT}^{-1} K_{NT}, K_2) + e_{p_1+1} e_{p_1}^{\mathsf{T}} K_{p_1+1,p_1}^q + e_{p_1} e_{p_1+1}^{\mathsf{T}} K_{p_1,p_1+1}^q + e_{p_2-1} e_{p_2}^{\mathsf{T}} K_{p_2-1,p_2}^q + e_{p_2} e_{p_2-1}^{\mathsf{T}} K_{p_2,p_2-1}^q.$$

Moreover, $\widetilde{K}_{NT}^{-1} K_{NT} = (I + K_{NT} D_{NT}^{(1)})^{-1} K_{NT} = (\Sigma_{NT} + D_{NT}^{(1)})^{-1}$, where $\Sigma_{NT} = K_{NT}^{-1}$ is the covariance of the random vector associated with $\mathcal{B}^{NT}$. Thus, $K^*$ in equation 18 and $(\Sigma^q)^{-1}$ are identical, except in their second diagonal blocks, as required. Furthermore, in order for the subgraph associated with $\widetilde{K}_{NT}$ to be a tree the entries in $\Sigma^q$ needs to be such that it matches the correlation factorization propert of a tree-subgraph. Using similar arguments, we can handle multiple internal blocks with block cut vertices that are not adjacent to leaf nodes. In the case where blocks have leaf nodes, we can combine the construction above with the construction in (Katiyar et al., 2019, Theorem 1) for tree structured graphical models. Combining these two, we can show that we can choose a decomposition $D = D_1 + D_2$ such that (a) the structure is arbitrarily different inside blocks, and (b) the block cut vertices are preserved (i.e., same as the ones in $G$), except they may be swapped with a neighboring leaf. $\qquad \square$