# OpenReview forum: "Robust Model Selection of Gaussian Graphical Models"
_TMLR — Accepted by TMLR_

### Review · Reviewer_nXhJ · 2024-11-20

**Summary Of Contributions:**

The paper considers the _robust model selection problem_ in the context of Gaussian graphical models (GGM).  Model selection in this context refers to the statistical problem of recovery of the sparsity pattern (an adjacency matrix) of the signal matrix of the GGM.  The word ``robust'' refers to a twist on the basic model selection problem in which Gaussian noise is added, independently to each variable.  The distribution of the noise for each variable need not be identically distributed.

The primary contributions of the paper toward this problem are
(i) a non-identifiability result -- namely, that there is a notion of equivalence on graphs that governs the extent to which sparsity patterns are recoverable;
(ii) an algorithm -- NoMAD for estimating the equivalence class of the underlying graph structure of the model from samples;
(iii) a theoretical accuracy analysis of NoMAD in both the population and finite-sample regimes.

**Audience:**

Yes

**Broader Impact Concerns:**

No ethical concerns.

**Claims And Evidence:**

Yes

**Requested Changes:**

Critical requests/questions:

1.) Theorem 3.1 seems to say that for a graph $G$, there is at least one graph $H$ in the equivalence class of $G$ whose noisy observations have the same distribution as those for $G$.  Isn't this trivial as stated?  After all, the equivalence class of $G$ contains $G$ itself.  Thus, either Theorem 3.1 is trivial, or needs to be revised.

2.) It seems like the statement at the top of page 6 (``... we know that any graph is only identifiable upto the equivalence relation in Definition 3.2'') is not an accurate interpretation of Theorem 3.1.  In particular, in order for this to be the correct interpretation, Theorem 3.1 would have to say that its conclusions hold _for all $H \in [G]$_.  As it is stated, it leaves open the possibility that there is potential for confusion among only some subset of elements of $[G]$.  This should be reworded.

3.) On page 5, it is stated that any graph whose AST is equal to $T_{AST}(G)$ is in $[G]$, and hence the authors will use $T_{AST}(G)$ to denote $[G]$.  I understand that the first statement is correct.  However, it is not clear to me that every element $H$ of $[G]$ has the same AST.  It is true, however, that all elements of $[G]$ have isomorphic ASTs.


4.) Given issue 2, I think that it would be appropriate to clarify in the introduction whether or not the present work completely solves the problem.  My understanding is that the proposed algorithm correctly identifies the equivalence class of the graph $G$, but the non-identifiability result (Theorem 3.1) does not imply that this is the limit of what can be identified.


Non-critical requested changes:

1.) There is a sentence fragment at the end of the first page.  Please correct it.

2.) The caption of Theorem 3.1 seems strange.  Would it be better to call it "Non-identifiability"?  If so, please do so.

3.) In Fact 1, page 6, $\rho$ is not defined yet.

4.) In the appendix, page 16, it is stated that ``... in this phase, it returns a collection $\mathfrak{B}$ of vertex triplets...''.  I believe that $\mathfrak{B}$ should be a collection of sets of vertex triplets, right?  This should be corrected, if so.

5.) In Theorem 5.3, the constant $C$ is uniform over all $\tau$, right?  The ordering of the quantifiers makes this slightly confusing.

**Strengths And Weaknesses:**

Strengths:

1.) The paper gets close to a tight characterization of the limits of robust model selection in Gaussian graphical models.

2.) An algorithm for estimating the equivalence class of the underlying graph structure is provided and explored theoretically.

3.) Modulo some issues explained below, the results seem to be more or less correct.

4.) Experiments comparing the proposed algorithm with an existing one are performed.  Both a real and a synthetic dataset are used.

Weaknesses:

1.) Certain crucial statements may need revision in order to meet the correctness and clarity standards for acceptance, and there are several typos.  Please see the "Requested changes" section.

---

> ### Author Response · Authors · 2025-01-07
> **Clarifying the Unidentifiability Result: Revisions and Implications**
>
> ## Revised Unidentifiability Result
>
> We thank the reviewer for their insightful feedback. The way our identifiability result was stated made it sound weaker, and we have now rephrased it in the revised manuscript. Below is the rephrased result, and we provide some commentary on its implications which we believe answers your questions.
>
>  > Fix a covariance matrix $\Sigma^*$ whose conditional independence structure is given by the graph $G$. Suppose we are given a noisy covariance matrix $\Sigma^{o}=\Sigma^*+D$ where $D=\textrm{diag} (D_{11},\ldots,D_{pp})\geq 0$. Then, for any $H\in [G]$ where $H \neq G$, there exists matrices $\Sigma_H, D_H$ such that  $\Sigma^{o} = \Sigma_H+D_H$,  $D_{H}$ is a diagonal matrix with non-negative entries, and the sparsity pattern of $(\Sigma_H)^{-1}$ is described by $H$.
>
>
>
> - From the above statement, it is clear that for any graph $G$, the set defined (Definition 3.2) as $[G]$ contains all graphs that can be confused with $G$ under noisy observation. Therefore, this is a true identifiability result in this sense.
>
> **Recovery of $[G]$ from $\mathcal{T}_{AST}$**: Finally, notice that using the operations described in Page 5, given a $\mathcal{T}_{AST}$ which equals to $\mathcal{T}\_{AST}(G)$ one can recover all the graphs in $[G]$.
>
> We have chosen to not identify $[G]$ with $\mathcal{T\}_{AST}(G)$, as this causes confusion. Thanks for pointing this out.
> ## Addressing Non-Critical Comments
>
> - In the appendix, page 16, $B$ should be a collection of sets of vertex triplets. We have corrected this typo.
> - In Theorem 5.3, the constant $C$ is uniform over all $\tau$. We have added a note to clarify this.
> -  The quantity $\rho$ has now been defined before its first appearance in Fact 1.
>
> **Page Break Issue:** This has been resolved in the revised document.

---

### Review · Reviewer_86eb · 2024-12-05

**Summary Of Contributions:**

The authors investigate the problem of structure learning for Gaussian graphical models using noisy samples. While previous studies have primarily focused on tree-structured graphs, this work extends the analysis to a broader class of graphs. The authors first define and characterize an equivalence class for general graphs, establishing a framework for the problem. Building on this, they propose an ancestor-discovery-based algorithm capable of identifying the underlying graph structure up to the specified equivalence class. The correctness of the algorithm is demonstrated under both population and sample settings, highlighting its theoretical validity.  Numerical experiments have been conducted to show the practical applicability of the proposed algorithm.

**Audience:**

Yes

**Claims And Evidence:**

Yes

**Requested Changes:**

In addition to the weaknesses highlighted in the previous section, I observed that the paper abruptly ends on page 1, with page 3 immediately following. This suggests that a significant portion of the related work section may be missing. The authors should verify whether the correct version of the manuscript was submitted.

**Strengths And Weaknesses:**

**Strengths**

The authors make a novel contribution by establishing an equivalence class for general graphs, which serves as a foundation for their work. In particular, the introduction of the “articulated set tree” and its innovative use in developing an algorithm to uncover the underlying graph structure in Gaussian graphical models is interesting. The proposed NoMAD algorithm, along with its theoretical guarantees in the population setting, further underscores the novelty and significance of their approach.

**Weaknesses**

My main concerns are regarding the assumptions:
1. The proposed work relies on two key assumptions: strong faithfulness and strong ancestor consistency in the finite sample setting. However, the practicality of these assumptions is unclear. Uhler et al. (2013) demonstrated that the strong faithfulness assumption is highly restrictive, with a surprisingly large set of distributions failing to satisfy it, even for small graphs. This raises concerns about the practical applicability of the proposed algorithm. Could the authors comment on this limitation? Furthermore, a discussion on the applicability of the strong ancestor consistency assumption in finite sample settings would also be valuable. Does the proposed approach require prior knowledge of $\gamma$ and $\zeta$?
2. The experimental setup requires further clarification. The experiments appear to be conducted on small graphs, raising questions about the scalability of the algorithm. Additionally, could the authors explain the meaning of the x-axis and y-axis in the plots presented in the main paper? Regarding the noise matrix, how many diagonal entries were non-zero? Furthermore, does the number of non-zero corrupted entries impact the algorithm’s performance? Providing more details on these aspects would help better understand the evaluation framework and its implications.
3. The authors mention the following as a critic of the existing works
> unfortunately the conditional independence structure of the underlying distribution can be completely lost in general under such corruption; ..

Is the proposed method (and the underlying assumptions) robust to noise/corruption?

---

> ### Author Response · Authors · 2025-01-07
> **Practicality of Assumptions in Finite Sample Settings and Clarifications on Simulation Setup**
>
> ## Practicality of Assumptions in Finite Sample Settings:
>
> The reviewer’s comment regarding the practical applicability of the strong faithfulness and strong ancestor consistency assumptions in finite sample settings is well taken. However, this practicality concern is not unique to our algorithm. Several well-known algorithms rely on partial correlation testing for structure learning in the Gaussian setting, and need such assumptions. [1] discusses several such methods.
>
>
> To address these concerns in finite sample settings, practitioners use a range of methods including using domain knowledge to bound hyperparameters, (Bayesian) parameter tuning, and stability selection [2]. It is worth highlighting that our paper represents the first step toward addressing the robust model selection problem for general structures, and establishes the feasibility (and limits) of this challenging problem.  Relaxing such assumptions can be a fruitful avenue for future work. We will include a discussion about this in the final manuscript, including the below discussion on how one may use a technique like stability selection in the context of this problem:
>
> - A potential way to address this issue in finite sample settings is by leveraging stability selection, as introduced by [2]. Stability selection is particularly effective in high-dimensional settings (where $p\gg n $) and aims to provide a framework for model selection in structure learning problems while controlling the number of false discoveries. Let $\Gamma$ be the set of all hyperparameter pairs $\gamma$ and $\zeta$. Let $\hat{E}^{\lambda}$ be the edge set for a fixed $(\gamma, \zeta) \in \Gamma$ returned by our algorithm. Furthermore, let $\hat{\Pi}_K^{\lambda}= \mathbb{P}(K\subseteq \hat{E}^{\lambda}(I))$, where $I$ is a random subsample of the available dataset. Then, output edge set $\hat{E}^{ stable}$ is the set of indices $k$ such that such that the maximum value of $\hat{\Pi}_k^{\lambda}$ over all pairs $(\gamma, \zeta)$ in the set $\Gamma$ is greater than or equal to a threshold value $\pi\_{\rm thr}$.
>
> How to select the parameter region $\Gamma$ and threshold $\pi_{\rm thr}$ is a promising avenue for future research work. We will include this comment in our final manuscript.
>
> ## Clarifications on Experimental Setup:
>
> We want to emphasize that the primary goal of our experimental results was to provide initial proof-of-concept demonstrations the success of our algorithm. Considerations of scalability of our algorithm is crucial, and we discuss this in the manuscript. Some minor comments on the setup are addressed below:
>
> - The $x$-axis represents the maximum allowed diagonals, while the $y$-axis represents the 0-1 loss for recovering an equivalent graph.
> - In our simulation, we considered all entries of the diagonals to be non-zero. It is important to note that, in our problem setup, the number of non-zero entries is not known beforehand. Therefore, the number of non-zero entries does not affect the algorithm's performance.
>
> **Robustness of Proposed Method:** The proposed method is indeed robust to noise and corruption—this is, in fact, a key feature of our work. In Theorem 3.1, we establish the limits upto which a graph can be recovered under the noise model described in Section 3.1. Furthermore, Theorem 4.2 demonstrates that our algorithm successfully recovers the correct equivalence class.
>
> **Page Break Issue:** This has been resolved in the revised document.
>
>
> **References**:
> 1. Mathias Drton and Marloes H Maathuis. “Structure learning in graphical modeling”. In: _Annual Review of Statistics and Its Application_ 4 (2017), pp. 365–393.
>
> 2. Nicolai Meinshausen and Peter Buhlmann. “Stability selection”. In:_Journal of the Royal Statistical Society Series B: Statistical Methodology_ 72.4 (2010), pp. 417–473.

---

### Review · Reviewer_k1Ff · 2025-01-24

**Summary Of Contributions:**

The problem tackled by the paper is how to estimate the underlying graph structure of a Gaussian Graphical Model simply from data, and in the presence of noise. In other words, it is about estimating the sparsity structure of the covariance matrix from data of the form Y = X + D, where D is a diagonal of noise - where each noise component has a potentially different variance or can be 0.
After giving some background presentation of the problem, the paper then introduces its main theoretical result (Theorem 3.1) which states that it is only possible to recover the graph structure up to an equivalence class in the presence of noise.
It then introduces the NoMAD algorithm, which aims at estimating this equivalence class. Note that both the theoretical result and the NoMAD algorithm do not assume a given structure of the graph to be estimated (unlike prior literature presented which assumed the structure to be a tree).
The paper finally provides some theoretical analysis of both the finite sample and asymptotic performance of NoMAD before showing some experimental results on generated data.

**Audience:**

Yes

**Claims And Evidence:**

Yes

**Requested Changes:**

See weaknesses section.

**Strengths And Weaknesses:**

Strengths:
- Paper and reasoning are well structured, explanation as clear and easy to follow
- Unidentifiability result as well as the NoMAD algorithm and its theoretical analysis as rigorous and sound, giving a strong grounding to the method
- Method presented opens a number of future research avenues
- Experimental results look promising


Weaknesses:
- I believe the main weakness of the paper is the fairly limited experimental section: the experiments presented are only on generated data, and the analysis of the results is not very detailed.
- Following on the previous point, the only benchmark method shown was Graphical Lasso, while it would have been a good reference to show how NoMAD performs compared to other sparse inverse covariance matrix estimation methods, e.g. [1]
- Another weakness: no code for reproducibility

[1] D. Bertsimas, J. Lamperski, J. Pauphilet, Certifiably Optimal Sparse Inverse Covariance Estimation

---

> ### Author Response · Authors · 2025-01-30
> **Regarding Experiments**
>
> The primary goal of our paper is to consider the robust model selection problem from a theoretical perspective. Toward this, we establish strong unidentifiability results and propose a method that provably recovers the graph up to the identifiability conditions we establish. The goal of our experimental results was to provide an initial proof-of-concept demonstration of our algorithm's success.  For this purpose, we selected GLASSO to illustrate how a standard algorithm completely fails in model selection in the proposed problem setup. We also demonstrate results in purely synthetic and quasi-synthetic (i.e., data generated from realistic systems used by practitioners -- the IEEE bus system) settings as a proof-of-concept.  Promising future work in this area includes the design and analysis of practical algorithms that build on our theory -- we discuss several concrete steps toward this in Section 7. We believe that it would be more appropriate to exhaustively evaluate such an algorithm empirically, while also comparing against various baselines like the one that the reviewer mentions.

---

### Decision · Action_Editor_bhA8 · 2025-03-16

**Recommendation:** Accept as is

**Comment:**

After discussion, all reviewers are in favour of acceptance as-is.

**Audience:**

Yes, this is clearly within the TMLR scope.

**Claims And Evidence:**

This submission studies robust recovery of GGMs, extending existing results on robust recovery of Gaussian tree models to a general class of graphs. The main result is a characterization of which graphs can be robustly recovered and an algorithm for recovery. The main concerns raised reflect the strength of the assumptions (e.g. strong faithfulness), but these are appropriate assumptions and although strong, this is a primarily theoretical paper that makes progress on a difficult problem.